# FedMGP: Personalized Federated Learning with Multi-Group Text-Visual Prompts

Weihao Bo[1], Yanpeng Sun[2]*, Yu Wang[3], Xinyu Zhang[4], Zechao Li[1]

[1]Nanjing University of Science and Technology
[2]National University of Singapore
[3]Baidu VIS
[4]University of Auckland

## Abstract

In this paper, we introduce **FedMGP**, a new paradigm for personalized federated prompt learning in vision-language models (VLMs). Existing federated prompt learning (FPL) methods often rely on a single, text-only prompt representation, which leads to client-specific overfitting and unstable aggregation under heterogeneous data distributions. Toward this end, FedMGP equips each client with *multiple groups* of paired textual and visual prompts, enabling the model to capture diverse, fine-grained semantic and instance-level cues. A diversity loss is introduced to drive each prompt group to specialize in distinct and complementary semantic aspects, ensuring that the groups collectively cover a broader range of local characteristics. During communication, FedMGP employs a dynamic prompt aggregation strategy based on similarity-guided probabilistic sampling: each client computes the cosine similarity between its prompt groups and the global prompts from the previous round, then samples *s* groups via a softmax-weighted distribution. This soft selection mechanism preferentially aggregates semantically aligned knowledge while still enabling exploration of underrepresented patternseffectively balancing the preservation of common knowledge with client-specific features. Notably, FedMGP maintains parameter efficiency by redistributing a fixed prompt capacity across multiple groups, achieving state-of-the-art performance with the lowest communication parameters (5.1k) among all federated prompt learning methods. Theoretical analysis shows that our dynamic aggregation strategy promotes robust global representation learning by reinforcing shared semantics while suppressing client-specific noise. Extensive experiments demonstrate that FedMGP consistently outperforms prior approaches in both personalization and domain generalization across diverse federated vision-language benchmarks. The code will be released on `https://github.com/weihao-bo/FedMGP.git`.

## 1 Introduction

Large-scale vision-language models (VLMs) have demonstrated impressive performance across a wide range of multimodal tasks [1, 31, 15, 43, 19–21, 42]. As these models are increasingly deployed in privacy-sensitive and decentralized environmentsincluding healthcare, mobile devices, and industrial systemsthere is a growing need to adapt them privately without direct access to raw data [5]. In such settings, data remains local, and client distributions are often highly heterogeneous [16, 8]. To fully utilize local data, personalized federated learning (PFL) [12, 50] has emerged as an effective framework for adapting shared models across clients with non-identical data, while preserving privacy. In parallel, prompt-based tuning has shown great promise for parameter-efficient adaptation of frozen VLMs. The integration of these two ideas has led to the rise of federated prompt learning (FPL)a lightweight and scalable approach to adapting VLMs in federated settings[17, 23].

---

*Corresponding author.

39th Conference on Neural Information Processing Systems (NeurIPS 2025).

Despite its potential, existing FPL methods face key limitations. Most approaches rely solely on textual prompts, which encode static class-level semantics. While efficient, these prompts lack the expressiveness to capture personalized visual cues specific to each client, limiting their ability to handle diverse or complex inputs. Furthermore, many methods adopt a local-global prompt framework [27, 9, 37], in which each client maintains a local prompt and contributes a single global prompt for aggregation. This framework introduces two critical problems: (1) A single prompt per client is often insufficient to capture the diversity of local dataespecially when multiple semantic concepts or visual styles coexist within a client. (2) Aggregating one prompt per client leads to biased global representations, as the shared prompt tends to overfit to dominant local patterns while overlooking less frequent but informative ones from other clients. Together, these issues undermine both local personalization and cross-client generalization, particularly under severe data heterogeneity.

To overcome these limitations, we propose Personalized **Fed**erated Learning via **M**ulti-Group Text-Visual **P**rompt (**FedMGP**), a new paradigm for personalized federated adaptation of vision-language models. Each client in FedMGP maintains multiple paired groups of textual and visual prompts, where each group captures distinct semantic and instance-level characteristics of the local data. To ensure prompt groups specialize in different aspects, we introduce a diversity loss that encourages representational separation within each client. For server aggregation, we develop a dynamic prompt selection strategy based on the similarity between local prompt groups and the global prompt from the previous round, ensuring that semantically aligned groups are more likely to be selected, while still allowing exploration of less dominant patterns. This balanced approach reinforces common cross-client patterns while suppressing client-specific noise.

FedMGP is both parameter-efficient and communication-aware: with the lowest communication parameters (5.1k) among all federated prompt learning methods, it achieves state-of-the-art performance while distributing a fixed prompt capacity across multiple groups. Empirical results across various heterogeneous data settings, including pathological non-IID, Dirichlet distribution, and domain generalization, demonstrate that FedMGP successfully balances personalization accuracy on local client data with generalization capability to unseen domains.

## 2 Related Work

### 2.1 Prompt Learning for Vision-Language Models

Vision-language models (VLMs) like CLIP [43] have demonstrated strong zero-shot capabilities through contrastive learning on massive image-text pairs [56, 55, 49, 11, 48]. To efficiently adapt these models to downstream tasks, prompt learning introduces a small set of learnable parameters while keeping the original model weights frozen [60, 30, 58]. Various prompt learning approaches have been proposed, including enriching text representations through class-related descriptions [53, 35], additional descriptive sentences [40, 57, 44], external knowledge [22], and visual annotations [46, 45]. As highlighted in our introduction, recent methods have begun addressing the critical balance between fitting to seen classes and maintaining generalization capabilities to unseen classes. For instance, CoCoOp [59] introduces instance-conditioned prompts to capture fine-grained visual cues while preserving general knowledge, and ProGrad [61] proposes prompt alignment gradients to maintain the model's inherent knowledge. However, these methods predominantly operate in centralized settings with direct access to all training data, overlooking privacy concerns and the challenges of heterogeneous data distributions across multiple clientscritical limitations that necessitate new frameworks for privacy-preserving, distributed adaptation of VLMs [29].

### 2.2 Federated Prompt Learning

Federated Prompt Learning (FPL) [17, 16, 51, 32] combines prompt learning with federated learning [34, 3, 8, 52, 28] to enable privacy-preserving adaptation of vision-language models across distributed environments. PromptFL [17] pioneered this approach by integrating prompt learning into federated frameworks with theoretical convergence guarantees. To address client heterogeneity, several researchers [27, 37, 16] developed local-global paradigms where clients maintain personalized prompts while contributing to shared global prompts. This approach improves local performance but often compromises generalization under non-IID data distributions. Recent work [9] attempted to balance personalization and generalization through additional constraints. Despite progress, ex-

isting FPL methods have two key limitations. First, they rely solely on textual prompts, missing crucial visual cues needed for robust multimodal adaptation. Second, they lack effective prompt learning strategies and aggregation mechanisms tailored specifically for federated settings that can simultaneously maintain personalization while enhancing cross-client generalization.

# 3 Method

In this paper, we introduces FedMGP, a novel approach designed to address data heterogeneity and model stability challenges in federated learning. We first present the fundamentals of federated prompt learning (Section 3.1), including the core concepts of prompt learning and its application in federated settings. Then, we elaborate on two key mechanisms of FedMGP: the multimodal prompt co-learning mechanism (Section 3.2.1), which enhances representation capabilities through the synergistic interaction between text and visual prompts, and the dynamic prompt aggregation strategy (Section 3.2.2), which effectively balances global knowledge sharing with local feature preservation.

## 3.1 Preliminary: Federated Prompt Learning

Prompt learning is a parameter-efficient strategy for adapting large pre-trained Vision-Language Models (VLMs), such as CLIP [43], to diverse downstream tasks. It introduces a small set of learnable parameters called "prompts" while keeping the VLM's encoder weights frozen. These learnable prompt vectors are combined with class name embeddings to create class-specific textual prompts that effectively adapt the model to downstream tasks.

A VLM typically consists of an image encoder $f(\cdot)$ and a text encoder $g(\cdot)$. The core workflow involves: (1) processing an input image through the image encoder to obtain visual features, (2) processing text prompts through the text encoder to obtain textual features, and (3) computing similarity scores between these features to determine class probabilities. The key prediction formula is:

$$p(\hat{y} = k|x; p_t) = \frac{\exp(\text{sim}(f(x), g(t_k))/\tau)}{\sum_{j=1}^{K} \exp(\text{sim}(f(x), g(t_j))/\tau)}. \tag{1}$$

where $t_k = \{p_t, c_k\}$ represents the text input formed by concatenating the learnable text prompt $p^t$ with the embedding of class name $c_k$. Here, $\text{sim}(\cdot, \cdot)$ represents cosine similarity, $K$ is the number of classes, and $\tau$ is a temperature scaling factor.

Federated Prompt Learning (FPL) extends this approach to distributed settings where multiple clients collaborate without sharing their raw data. The federated training process follows a cyclic pattern: (1) The server distributes the current global prompt to selected clients; (2) Clients perform local updates using their private data; (3) Updated local prompts are sent back to the server; (4) The server aggregates these local prompts to form an improved global prompt. This process repeats for multiple communication rounds, gradually refining the global prompt to work well across all clients.

Despite its privacy-preserving benefits, standard FPL faces significant challenges with heterogeneous client data distributions. Client-specific optimization may lead to overfitting to local patterns, while naive aggregation methods like FedAvg [34] often struggle to preserve client-specific knowledge while extracting common patterns. Our proposed FedMGP framework specifically addresses these limitations through a multi-group prompt architecture and dynamic prompt aggregation strategy.

## 3.2 FedMGP:Federated Learning via Multi-Group Text-Visual Prompt

To address the fundamental limitations of existing federated prompt learning methods, particularly their reliance on single text-only prompts and vulnerability to client-specific overfitting, we propose FedMGP. (The complete pseudocode can be found in appendix A.) As illustrated in figure 1, our framework introduces a novel multi-group mechanism that enhances both prompt diversity and robustness through complementary prompt groups. For each client in the federation of $N$ clients, we define a set of prompts $P = \{p_{t,1}, \ldots, p_{t,G}, p_{v,1}, \ldots, p_{v,G}\}$, where $p_{t,j}$ represents the $j$-th text

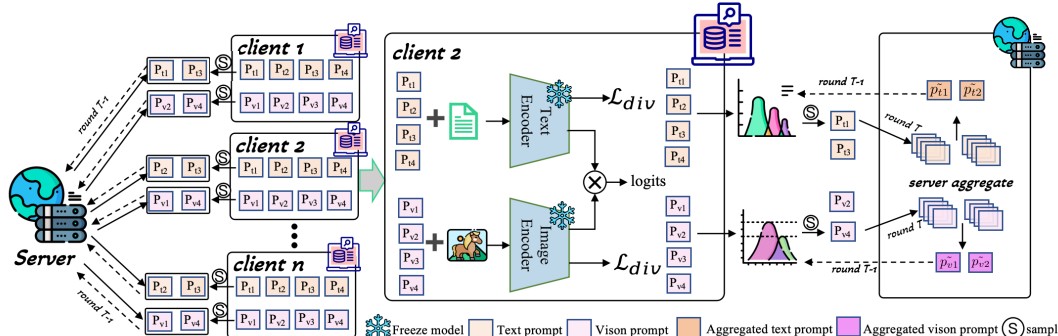

Figure 1: Overview of FedMGP: The left portion shows the server distributing global prompts to clients; the middle portion illustrates the multi-group text-visual prompt co-learning mechanism within each client; and the right portion demonstrates the dynamic prompt aggregation strategy across communication rounds.

prompt and $p_{v,j}$ represents the $j$-th visual prompt, with $G$ being the number of prompt groups. We use $\tilde{P}$ to denote the global aggregated prompts at the server and T denote the communication round.

This design offers two significant advantages. First, integrating visual and textual modalities enriches contextual representation, capturing instance-specific information more comprehensively than static class names. Second, distributing knowledge across multiple specialized prompt units enhances aggregation robustnesseven if certain prompt groups overfit to local distributions, others may capture generalizable patterns, significantly improving model adaptability under heterogeneous data distributions without increasing the total parameter count.

### 3.2.1 Multimodal Prompt Co-learning Mechanism

For any local client, the multi-group prompt learning process operates as follows. During training, for each group $j$, the text prompt $p_{t,j}$ is concatenated with the class embedding $c_k$ to form $t_{k,j} = \{p_{t,j}, c_k\}$, which is fed into a text encoder $g(\cdot)$. Simultaneously, the image $x$ is combined with the visual prompt $p_{v,j}$ to form $v_j = \{x, p_{v,j}\}$, which is passed through the image encoder $f(\cdot)$. The predictive probability for class $k$ is computed based on the similarity between the corresponding text and visual features:

$$p(\hat{y} = k \mid x; p_{t,j}, p_{v,j}) = \frac{\exp(\mathrm{sim}(f(v_j), g(t_{k,j}))/\tau)}{\sum_{l=1}^{K} \exp(\mathrm{sim}(f(v_j), g(t_{l,j}))/\tau)}. \tag{2}$$

The classification loss is defined as the average cross-entropy across all $G$ prompt groups:

$$\mathcal{L}_{\mathrm{CE}} = \frac{1}{G} \sum_{j=1}^{G} -\log p(\hat{y} = y \mid x; p_{t,j}, p_{v,j}) \tag{3}$$

To ensure that different prompt groups capture diverse semantic perspectives, we introduce a diversity loss that minimizes the cosine similarity between group-wise features within the same modality:

$$\mathcal{L}_{\mathrm{div}} = \sum_{k=1}^{K} \sum_{j \neq j'} (1 - \cos(g(t_{k,j}), g(t_{k,j'}))) + \sum_{j \neq j'} (1 - \cos(f(v_j), f(v_{j'}))) \tag{4}$$

This encourages each group to specialize in different aspects of the input, thereby reducing redundancy and enhancing representational richness. The overall training objective combines both losses:

$$\mathcal{L} = \mathcal{L}_{\mathrm{CE}} + \lambda \cdot \mathcal{L}_{\mathrm{div}} \tag{5}$$

At inference time, we leverage all prompt groups by computing predictions independently for each group and averaging the resulting logits:

$$p(\hat{y} = k \mid x) = \frac{1}{G} \sum_{j=1}^{G} p(\hat{y} = k \mid x; p_{t,j}, p_{v,j}) \tag{6}$$

This group-wise ensemble enhances robustness by aggregating complementary semantic views, improving prediction stability across heterogeneous inputs while incurring minimal overhead.

### 3.2.2 Dynamic Prompt Aggregation Strategy

To address the aggregation instability issues prevalent in traditional federated learning, FedMGP employs a novel dynamic prompt aggregation strategy, as illustrated in the right portion of Figure 1. This approach is based on a fundamental insight: each prompt can be conceptually decomposed into information that is common across clients (global knowledge) and information that is unique to a specific client (local knowledge). Therefore, we propose this dynamic aggregation mechanism that adaptively balances the preservation of global knowledge with the exploration of client-specific features. In Appendix F, we provide theoretical analysis demonstrating the superiority of our dynamic aggregation strategy over both full prompt aggregation methods[17, 34] and explicit global-local paradigms [27, 9, 37], offering formal justification for our approach.

In this section, we use $P$ without subscripts to refer to the entire set of prompts, while $P_j = \{p_{t,j}, p_{v,j}\}$ refers to the $j$-th group within that set. The global aggregated prompts are denoted by $\tilde{P}$, where $\tilde{P}$ consists of top-$s$ selected prompt groups.

In each communication round, our strategy dynamically selects a subset of top-$s$ prompts from each client for aggregation, where $s \leq G$ and $G$ is the total number of prompt groups. When $s = G$, our method reduces to standard FedAvg. By selecting only the most relevant prompts, we focus the aggregation on shared knowledge while preserving client specificity. The key steps of our dynamic aggregation strategy are as follows:

For communication round $T$, we first compute the cosine similarity between each client's local prompts and the global prompts from the previous round. This process is performed separately for text and visual prompts, but follows the same procedure. For each local prompt group $P_j^T$ and its corresponding global prompt $\tilde{P}_i^{T-1}$ (where $i \in \{1, 2, \ldots, s\}$ indexes the top-$s$ selected groups):

$$\text{sim}(P_j, \tilde{P}^{T-1}) = \sum_{i=1}^{s} \frac{P_j^T \cdot \tilde{P}_i^{T-1}}{||P_j^T|| \cdot ||\tilde{P}_i^{T-1}||} \tag{7}$$

To avoid overly deterministic selection that might lead to prompt homogenization, we convert these similarity scores into selection probabilities using a softmax function with temperature parameter $\tau$:

$$\text{prob}(P_j^T) = \frac{\exp(\text{sim}(P_j^T, \tilde{P}^{T-1})/\tau)}{\sum_{j=1}^{G} \exp(\text{sim}(P_j^T, \tilde{P}^{T-1})/\tau)} \tag{8}$$

Based on these probabilities, we sample $s$ prompt groups from each client. For the first communication round ($T = 1$), when no previous global prompts exist, we employ random selection to establish initial diversity, as described in Appendix A.

After selecting the top-$s$ most relevant prompt groups from each client, the server aggregates these prompts across all participating clients to form the updated global prompts for round $T$. For the $i$-th selected prompt group:

$$\tilde{P}_i^T = \sum_{c \in C_T} \frac{n_c}{\sum_{c' \in C_T} n_{c'}} P_{i,c}^T, \tag{9}$$

where $C_T$ represents the set of clients participating in round $T$, $n_c$ denotes the number of samples at client c and $P_i^c$ represents the $i$-th selected prompt group from client $c$.

This dynamic prompt aggregation strategy offers several key advantages. First, by favoring prompts with higher similarity to previous global prompts, we effectively filter out client-specific idiosyncrasies that might arise from local data distribution peculiarities. Second, the dynamic nature of our selection process prevents premature convergence to a static set of prompts, allowing the model to continually explore the prompt space and adapt to evolving patterns in the data. Third, this approach naturally balances the preservation of common knowledge with the exploration of diverse prompt configurations, leading to more robust federated learning.

After aggregation, the server distributes the updated global prompts back to the clients for the next round, continuing this process for multiple communication rounds to gradually refine the global prompts to work well across all clients.

# 4   Experiment

In this section, we conduct comprehensive experiments to validate the dual capabilities of FedMGP: (1) maintaining strong personalization for individual clients while achieving robust cross-client generalization, and (2) demonstrating superior performance across various heterogeneous data distributions. Our evaluation spans multiple scenarios including non-IID data partitions and Dirichlet distributions with varying concentration parameters, demonstrating FedMGP's effectiveness in addressing the fundamental challenges of federated learning with prompt-based multimodal adaptation.

## 4.1   Experimental Setup

**Datasets and Data Heterogeneity.**To thoroughly evaluate FedMGP's dual capabilities of personalization and generalization across heterogeneous data distributions, we design experiments with three distinct scenarios. First, following [9, 16], we select nine diverse datasets to assess base-to-novel class generalization: Caltech101 [13] for general object classification; OxfordPets [38], Flowers102 [36], Food101 [4], Stanford Cars [25], and FGVC Aircraft [33] for fine-grained classification; DTD [7] for texture classification; UCF101 [47] for action recognition; and SUN397 [54] for scene recognition. We create a pathological non-IID setting by equally splitting each dataset into base and novel classes, then assigning non-overlapping base classes to different clients. Each client's model is trained on local classes and evaluated on three test sets: local classes (personalization), base classes seen by other clients (cross-client knowledge transfer), and novel classes unseen during training (generalization to new concepts).Second, to evaluate personalization under label distribution shift, we employ CIFAR-10 and CIFAR-100 [26], partitioning data among clients using Dirichlet distribution $\text{Dir}(\alpha)$ with varying concentration parameters. This creates realistic heterogeneity where clients possess varying class proportions, allowing us to examine how effectively FedMGP's multi-group prompt mechanism adapts to imbalanced class distributions.Third, to assess performance under both feature and label distribution shifts, we test FedMGP on multi-domain datasets: DomainNet [39] with six distinct visual domains and Office-Caltech10 [14] with four domains. This evaluates how effectively our text-visual prompt co-learning bridges domain gaps while maintaining local specialization. Comprehensive dataset details are provided in Appendix C.1.

**Implementation Details.**   To ensure fair comparison with existing methods, we establish a unified experimental framework by re-implementing all baseline approaches using their official code repositories under identical settings. Specifically, we adopt ViT-B/16 [10] as the backbone for all methods. For base-to-novel generalization experiments, we set communication rounds $T = 10$ with 100% client participation rate, local epochs $E = 2$, and use 16-shot samples per class. For CIFAR-10 and CIFAR-

Table 2: Results on CIFAR10 and CIFAR100 with label shift with Dir partition($\alpha = 0.5$) into 100 clients.

| Methods | CIFAR10 | CIFAR100 |
|---|---|---|
| PromptFL [17] | 91.36 | 72.04 |
| FedOTP [27] | 94.73 | 75.15 |
| FedTPG [41] | 92.44 | 74.39 |
| FedPGP [9] | 92.41 | 74.11 |
| PromptFolio [37] | 93.33 | 74.14 |
| FedMGP | **95.48** | **75.39** |

100 experiments, we simulate a realistic federated environment with $\text{Dir}(\alpha = 0.5)$ distribution across 100 clients, with 10% client participation rate per round, utilizing the full training dataset. All models are trained using stochastic gradient descent (SGD) with an initial learning rate of 0.001

Table 1: Accuracy comparison (%) on clients' local accuracy and generalization.

**(a) Average over 9 datasets.**

| Methods | Local | Base | Novel | CM |
|---|---|---|---|---|
| PromptFL [17] | 71.19 | 71.70 | 71.46 | 71.31 |
| FedOTP [27] | 92.53 | 16.84 | 31.66 | 57.10 |
| FedTPG [41] | 71.62 | 71.91 | 68.32 | 70.66 |
| FedPGP [9] | 84.32 | 72.45 | 68.97 | 77.42 |
| PromptFolio [37] | 96.02 | 39.75 | 51.02 | 70.29 |
| FedMGP | 93.17 | 68.49 | 72.99 | **81.85** |

**(b) OxfordPets.**

| Methods | Local | Base | Novel | CM |
|---|---|---|---|---|
| PromptFL [17] | 89.77 | 90.01 | 97.20 | 91.62 |
| FedOTP [27] | 100.00 | 26.68 | 57.16 | 68.19 |
| FedTPG [41] | 94.24 | 94.31 | 96.64 | 94.85 |
| FedPGP [9] | 96.20 | 95.01 | 96.89 | 96.07 |
| PromptFolio [37] | 99.90 | 66.23 | 83.38 | 86.86 |
| FedMGP | 97.15 | 93.83 | 97.04 | **96.28** |

**(c) Flowers102.**

| Methods | Local | Base | Novel | CM |
|---|---|---|---|---|
| PromptFL [17] | 70.33 | 71.79 | 75.39 | 71.94 |
| FedOTP [27] | 99.73 | 13.06 | 21.51 | 57.99 |
| FedTPG [41] | 79.43 | 78.92 | 73.26 | 77.71 |
| FedPGP [9] | 91.83 | 80.22 | 68.46 | 82.85 |
| PromptFolio [37] | 99.82 | 27.36 | 39.34 | 66.05 |
| FedMGP | 98.41 | 70.06 | 74.71 | **85.36** |

**(d) DTD.**

| Methods | Local | Base | Novel | CM |
|---|---|---|---|---|
| PromptFL [17] | 55.32 | 57.06 | 44.32 | 52.60 |
| FedOTP [27] | 96.44 | 20.06 | 41.23 | 61.71 |
| FedTPG [41] | 56.90 | 59.26 | 40.46 | 52.49 |
| FedPGP [9] | 78.47 | 67.22 | 50.93 | 68.21 |
| PromptFolio [37] | 97.18 | 26.53 | 37.39 | 64.11 |
| FedMGP | 92.87 | 53.60 | 55.62 | **73.73** |

**(e) Caltech101.**

| Methods | Local | Base | Novel | CM |
|---|---|---|---|---|
| PromptFL [17] | 94.16 | 95.35 | 94.98 | 94.66 |
| FedOTP [27] | 99.96 | 28.28 | 62.26 | 69.43 |
| FedTPG [41] | 96.17 | 97.16 | 91.92 | 95.32 |
| FedPGP [9] | 96.91 | 97.35 | 94.37 | 96.37 |
| PromptFolio [37] | 99.79 | 73.69 | 81.10 | 88.50 |
| FedMGP | 99.47 | 96.02 | 93.61 | **97.13** |

**(f) Food101.**

| Methods | Local | Base | Novel | CM |
|---|---|---|---|---|
| PromptFL [17] | 89.75 | 89.79 | 90.86 | 90.04 |
| FedOTP [27] | 95.44 | 19.16 | 45.89 | 61.24 |
| FedTPG [41] | 90.36 | 90.42 | 91.78 | 90.73 |
| FedPGP [9] | 90.51 | 90.48 | 91.12 | 90.65 |
| PromptFolio [37] | 97.24 | 57.40 | 67.64 | 79.67 |
| FedMGP | 95.08 | 88.47 | 89.53 | **92.04** |

**(g) UCF101.**

| Methods | Local | Base | Novel | CM |
|---|---|---|---|---|
| PromptFL [17] | 77.08 | 76.94 | 70.36 | 75.29 |
| FedOTP [27] | 92.39 | 16.33 | 19.07 | 54.99 |
| FedTPG [41] | 76.22 | 75.96 | 72.09 | 75.10 |
| FedPGP [9] | 82.61 | 71.78 | 68.45 | 76.34 |
| PromptFolio [37] | 96.15 | 31.94 | 42.00 | 66.22 |
| FedMGP | 92.69 | 68.38 | 72.86 | **81.62** |

**(h) SUN397.**

| Methods | Local | Base | Novel | CM |
|---|---|---|---|---|
| PromptFL [17] | 76.25 | 76.20 | 75.68 | 76.09 |
| FedOTP [27] | 93.40 | 11.38 | 19.11 | 53.83 |
| FedTPG [41] | 73.72 | 73.71 | 75.17 | 74.08 |
| FedPGP [9] | 89.43 | 66.51 | 67.43 | 78.20 |
| PromptFolio [37] | 95.18 | 32.89 | 44.47 | 66.50 |
| FedMGP | 91.83 | 68.51 | 72.20 | **81.07** |

**(i) Stanford Cars.**

| Methods | Local | Base | Novel | CM |
|---|---|---|---|---|
| PromptFL [17] | 62.98 | 63.14 | 69.87 | 64.66 |
| FedOTP [27] | 91.06 | 9.32 | 10.62 | 50.49 |
| FedTPG [41] | 65.50 | 65.47 | 69.10 | 66.37 |
| FedPGP [9] | 85.37 | 57.63 | 60.19 | 72.13 |
| PromptFolio [37] | 96.44 | 29.43 | 46.77 | 66.28 |
| FedMGP | 92.61 | 56.48 | 71.19 | **77.80** |

**(j) FGVC Aircraft.**

| Methods | Local | Base | Novel | CM |
|---|---|---|---|---|
| PromptFL [17] | 25.03 | 25.03 | 24.48 | 24.89 |
| FedOTP [27] | 64.34 | 7.27 | 8.12 | 36.01 |
| FedTPG [41] | 12.00 | 12.00 | 4.50 | 9.27 |
| FedPGP [9] | 47.59 | 25.89 | 22.89 | 35.94 |
| PromptFolio [37] | 82.50 | 12.29 | 17.09 | 48.40 |
| FedMGP | 78.46 | 21.03 | 30.15 | **51.62** |

and a single-step learning rate scheduler. All other implementation specifics, including additional hyperparameter settings, optimization strategies, and evaluation protocols, are detailed in the appendix to ensure reproducibility. For more details, please refer to Appendix C.2.

**Baselines.** We compare FedMGP against state-of-the-art Federated Prompt Learning (FPL) methods, including PromptFL [17], FedOTP [27], FedTPG [41], FedPGP [9], and PromptFolio [37]. These baselines represent the full spectrum of existing FPL paradigms: from standard aggregation approaches to local-global frameworks and constrained local-global architectures. This comprehensive comparison allows us to evaluate how effectively FedMGP addresses the critical balance between personalization and generalization that many existing methods struggle to achieve, particularly under severe data heterogeneity.

## 4.2 Performance Evaluation

**Analysis of Base-to-Novel Generalization Results.** To comprehensively assess both personalization and generalization capabilities, we introduce a Combined Metric (CM) that balances local adaptation and cross-domain transfer. Following the approach in [17] for local accuracy evaluation and [60] for harmonized accuracy calculations, CM is computed as $CM = (Local + HM)/2$, where HM is the Harmonic Mean defined as $HM = 2 \times Base \times Novel/(Base + Novel)$. This metric

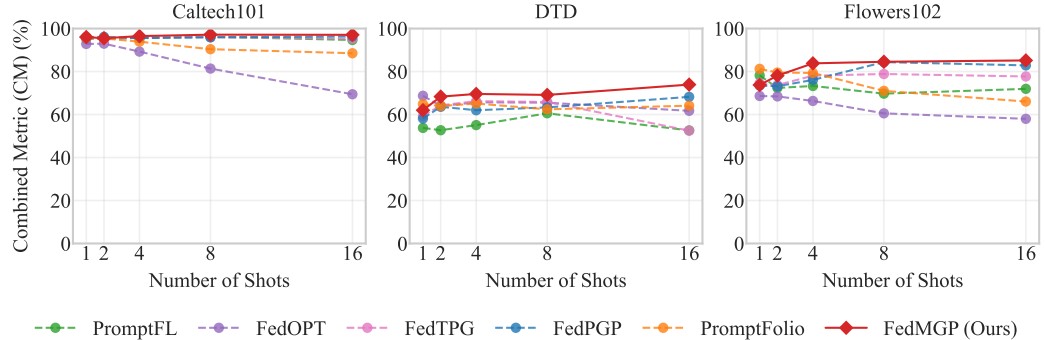

Figure 2: Few shot experiment from 1 to 16 shots

effectively quantifies a model's ability to simultaneously achieve personalization (measured by local accuracy) and generalization (measured by harmonized performance on base and novel classes). As shown in Table 1, FedMGP achieves the highest CM score (81.85%) averaged over all nine datasets, demonstrating superior overall performance while maintaining excellent balance between personalization (93.17% on local classes) and generalization (68.49% on base classes and 72.99% on novel classes). In contrast, methods like FedOTP and PromptFolio achieve exceptional local accuracy (92.53% and 96.02% respectively) but struggle with generalization to base classes (16.84% and 39.75%), indicating severe overfitting to local distributions. FedPGP, though more balanced, still falls short of FedMGP in comprehensive performance. These results confirm our analysis that existing approaches either excel at personalization at the expense of generalization or achieve moderate performance on both fronts without fully resolving the tension between these competing objectives.

Table 3: Parameter analysis of FedMGP and other state-of-the-art methods.

| Method | Trained | Communication | CM |
|---|---|---|---|
| PromptFL [17] | 8.2k | 8.2k | 80.27 |
| FedOTP [27] | 16.4k | 8.2k | 63.74 |
| FedTPG [41] | 4208.1k | 4208.1k | 82.37 |
| FedPGP [9] | 24.8k | 16.4k | 86.94 |
| PromptFolio [37] | 16.4k | 8.2k | 77.10 |
| FedMGP | 12.8k | 5.1k | **88.34** |

Table 4: Ablation study on prompt leangth($l$)

| Setting | Local | Base | Novel | CM |
|---|---|---|---|---|
| FedMGP ($l$=4) | 97.18 | 72.49 | 72.17 | 84.75 |
| FedMGP ($l$=8) | 98.05 | 64.00 | 64.91 | 81.25 |
| FedMGP ($l$=16) | 97.62 | 57.47 | 61.56 | 78.53 |
| FedMGP ($l$=2) | 96.92 | 73.23 | 74.65 | **85.43** |

**Performance on Label Distribution Shift.** We evaluate FedMGP's effectiveness in handling realistic federated learning scenarios with 100 clients following a Dirichlet distribution ($\alpha = 0.5$), which creates substantial heterogeneity in class distributions. As shown in Table 2, FedMGP consistently outperforms all baseline methods on both CIFAR-10 and CIFAR-100 datasets. The multi-group prompt mechanism effectively captures diverse client data patterns through text-visual prompt co-learning and similarity-based selection, enabling robust performance even under severe label imbalance. Notably, while other methods struggle with the increased complexity of CIFAR-100, FedMGP maintains its relative advantage, demonstrating strong scalability in federated learning with numerous clients and classes.

**Few-Shot Analysis.** Figure 2 demonstrates FedMGP's effectiveness across few-shot settings (1-16 shots per class). While FedMGP exhibits limitations in extreme 1-shot scenarios, it quickly surpasses competing methods with 2+ shots. This performance pattern aligns with our theoretical framework: in extremely limited data regimes, the multi-group mechanism struggles to effectively decompose knowledge into common and client-specific componentsa decomposition that is fundamental to our approach as described in Section 3.2.2. Specifically, with insufficient samples, prompt groups cannot effectively disentangle specialized representations nor establish robust text-visual correlations across client distributions. As sample size increases, FedMGP's dynamic prompt selection strategy activates its full potential, enabling superior cross-client knowledge transfer while preserving client-specific information. Detailed discussions on FedMGP's limitations and future research directions can be found in Appendix B.

Table 5: Ablation study on Prompt Groups($m$)

| Setting | Local | Base | Novel | CM |
|---|---|---|---|---|
| FedMGP ($m$=4) | 96.60 | 73.28 | 73.99 | 85.12 |
| FedMGP ($m$=3) | 89.95 | 77.68 | 74.48 | 83.00 |
| FedMGP ($m$=2) | 82.88 | 82.15 | 74.05 | 80.38 |
| FedMGP ($m$=1) | 78.85 | 79.68 | 70.67 | 76.88 |
| FedMGP ($m$=5) | 96.92 | 73.23 | 74.65 | **85.43** |

Table 6: Ablation study on Top-s.

| Setting | Local | Base | Novel | CM |
|---|---|---|---|---|
| FedMGP (Top-s=1) | 97.88 | 69.17 | 74.10 | 84.72 |
| FedMGP (Topk-s=3) | 92.93 | 76.88 | 74.85 | 84.39 |
| FedMGP (Topk-s=4) | 86.77 | 79.44 | 74.89 | 81.93 |
| FedMGP (Topk-s=2) | 96.92 | 73.23 | 74.65 | **85.43** |

**Parameter Efficiency.** Table 3 highlights FedMGP's remarkable communication efficiency (5.1k parameters)significantly lower than all competitors while achieving superior performance. This validates our core design: rather than increasing parameter count, FedMGP strategically distributes a fixed capacity across multiple specialized prompt groups, more effectively capturing diverse client data characteristics with minimal communication overhead. Additional evaluation like domain evaluation results are presented in Appendix D.

## 4.3 Ablation Study

To thoroughly understand FedMGP's design choices, we conduct extensive ablation studies examining key components including prompt length, number of prompt groups, top-s selection size, vision-text modality contributions, and diversity loss. For comprehensive evaluation and efficiency, all results are reported as the average performance across Caltech101, Flowers102, and DTD datasets, providing insights into FedMGP's optimal configuration.

**Impact of prompt length.** Table 4 reveals that increasing prompt length beyond $l$=2 causes consistent performance degradation. In heterogeneous federated environments, compact prompts excel by capturing essential semantic patterns without overfitting to client-specific details, enabling more effective knowledge sharing across diverse client distributions.

Table 7: Ablation study on the impact of vision and text prompt.

| Setting | Local | Base | Novel | CM |
|---|---|---|---|---|
| FedMGP (Vision Only) | 75.94 | 76.48 | 72.92 | 75.30 |
| FedMGP (Text Only) | 95.23 | 73.60 | 73.80 | 84.46 |
| FedMGP (Vision + Text) | 96.92 | 73.23 | 74.65 | **85.43** |

Table 8: Ablation study on $\mathcal{L}_{\text{div}}$.

| Setting | Local | Base | Novel | CM |
|---|---|---|---|---|
| FedMGP (w/o $\mathcal{L}_{\text{div}}$) | 94.53 | 72.97 | 72.48 | 83.63 |
| FedMGP ($\mathcal{L}_{\text{div}}$=2) | 95.78 | 74.88 | 74.98 | 85.35 |
| FedMGP ($\mathcal{L}_{\text{div}}$=5) | 96.35 | 73.50 | 74.31 | 85.13 |
| FedMGP ($\mathcal{L}_{\text{div}}$=10) | 95.78 | 73.09 | 74.35 | 84.75 |
| FedMGP ($\mathcal{L}_{\text{div}}$=1) | 96.92 | 73.23 | 74.65 | **85.43** |

**Effect of Group Number.** Table 5 shows that multiple prompt groups are crucial for FedMGP's effectiveness, with performance declining as group count decreases. Our results indicate that 5 groups achieves optimal performance, with additional groups likely offering diminishing returns relative to the increased parameter count. This validates our multi-group design which effectively balances personalization and generalization without rigid global-local separation.

**Selection Strategy Analysis.** Table 6 demonstrates how our dynamic prompt aggregation strategy navigates the critical personalization-generalization trade-off. With smaller selection size (Top-s=1), the model preserves client specificity but limits knowledge sharing, while larger selection size (Top-s=4) improves generalization but significantly compromises personalization. Top-s=2 emerges as the optimal balance point, effectively addressing the aggregation instability issues.

**Effect of Vision and Text Components.** Table 7 confirms the necessity of incorporating both vision and text components in FedMGP. Removing either modality leads to noticeable performance degradation, highlighting the complementary roles they play. While textual prompts capture high-level semantic categories, visual prompts provide fine-grained, instance-specific cues. Their joint contribution enables FedMGP to better represent diverse client data and facilitates more effective cross-client knowledge transfer. This finding supports the design choice of our multi-group text-visual prompt co-learning framework.

**Impact of diversity Loss.** Table 8 demonstrates the critical importance of diversity loss in FedMGP, with its removal causing a significant performance drop (CM decreases by 1.8%). This component

ensures effective separation between prompt groupsa fundamental mechanism we analyze in detail in Appendix E. Remarkably, performance remains stable across different weight values (1-10), confirming its insensitivity to hyperparameter settingsa significant advantage in federated environments with heterogeneous data distributions. Appendix E contains additional ablation studies on temperature parameters, diversity loss formulations, and other design choices.

## 4.4 Visual Analysis

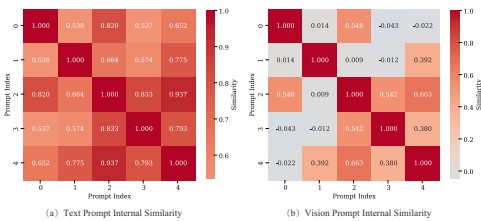

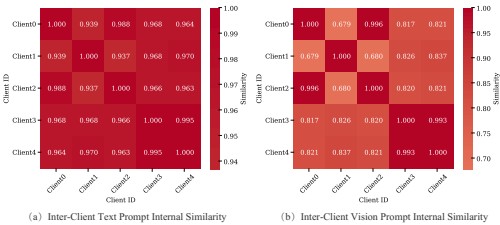

Figure 3: Intra-client prompt similarity visualization. (a) Text prompt similarity matrix showing moderate inter-group diversity. (b) Visual prompt similarity matrix showing higher intergroup diversity.

Figure 4: Inter-client prompt similarity visualization. (a) Text prompt similarity matrix showing high correlations (0.9-1.0). (b) Visual prompt similarity matrix showing moderate diversity (0.7-0.9).

To validate our diversity loss mechanism, we analyze internal prompt similarity patterns within a representative client after FedMGP training on Caltech101. Figure 3 presents similarity matrices for text and visual prompt groups, revealing distinct specialization. Text prompts show moderate inter-group correlations (0.5-0.8), maintaining shared linguistic patterns, while visual prompts exhibit significantly lower correlations (often near zero or negative), achieving superior diversification. This confirms that visual prompts capture more fine-grained, instance-specific features than text prompts. The diversity loss successfully encourages each prompt group to specialize in distinct patterns, enabling comprehensive local data coverage while supporting both personalization and cross-client generalization.

To further validate our dynamic aggregation mechanism, we examine inter-client prompt similarity patterns. Figure 4 shows that text prompts maintain consistently high correlations (0.9-1.0) across clients, preserving common semantic knowledge while avoiding the complete homogenization in PromptFL [17]. Visual prompts show moderate correlations (0.7-0.9), striking an optimal balance between knowledge transfer and client-specific adaptation. Unlike FedOPT's [27] global-local paradigm that often results in excessive divergence, our approach maintains sufficient similarity for knowledge sharing while preserving diversity for personalized learning. This confirms that FedMGP's dynamic aggregation effectively prevents over-homogenization and excessive divergence, achieving superior performance across heterogeneous client distributions.

## 5 Conclusion

This paper presents FedMGP, a novel federated learning paradigm that addresses the fundamental trade-off between personalization and generalization in existing federated prompt learning methods through multi-group text-visual prompt co-learning. The key innovations of FedMGP include: (1) leveraging multiple text-visual prompt pairs to overcome the limited expressiveness of single prompts, with each prompt group focusing on different semantic features; (2) introducing diversity loss to ensure representation separation between prompt groups, enhancing the model's expressive power; (3) designing a similarity-based dynamic prompt selection strategy that effectively balances shared knowledge and client-specific features. Extensive experiments demonstrate that FedMGP achieves superior balance between personalization and generalization capabilities across various heterogeneous data environments while maintaining minimal communication parameters. In future work, we will explore alternative regularization constraints and integrate category-specific linguistic information to further enhance diverse representations across prompt groups, while investigating more sophisticated text-visual prompt collaboration mechanisms to improve cross-modal alignment in federated settings.

**Acknowledge** This work was supported by National Natural Science Foundation of China (Grant No. 62425603) and Basic Research Program of Jiangsu Province (Grant No. BK20240011).

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

## Supplementary organization:

# A FedMGP ALGORITHM

---

**Algorithm 1** FEDMGP: Federated Learning via Multi-Group Text-Visual Prompt Co-Learning

---

**Inputs:** Communication rounds $T$, local epochs $R$, number of clients $N$, local datasets $\{D_c\}_{c=1}^N$, image encoder $f(\cdot)$, text encoder $g(\cdot)$, number of prompt groups $G$, top-$s$ size for aggregation, temperature $\tau$, diversity loss weight $\lambda$, learning rate $\eta$.

**Outputs:** Personalized multi-group prompts $\{P_c\}_{c=1}^N$, where $P_c = \{p_{t,1}, \ldots, p_{t,G}, p_{v,1}, \ldots, p_{v,G}\}$.

---

1: **Server Executes:**
2: Initialize global prompts $\tilde{P}^0 = \{\tilde{p}_{t,1}, \ldots, \tilde{p}_{t,G}, \tilde{p}_{v,1}, \ldots, \tilde{p}_{v,G}\}$.
3: **for** each client $c = 1, \ldots, N$ **do**
4:      Distribute copies: $P_c \leftarrow \tilde{P}^0$.
5: **end for**
6: **for** each communication round $T = 1, \ldots, T_{max}$ **do**
7:      Server selects a subset of clients $C_T$.
8:      **for** each client $c \in C_T$ **in parallel do**
9:          $P_c^T \leftarrow$ CLIENTUPDATE$(c, P_c, D_c, R, f, g, G, \tau, \lambda, \eta)$
10:      **end for**

                                             ▷ Dynamic prompt aggregation stage

11:      **for** each client $c \in C_T$ **do**
12:          **if** $T = 1$ **then**
13:              Select $s$ prompt groups randomly from $P_c^T$, denoted as $P_{c,selected}^T$
14:          **else**
15:              Compute similarity scores between client prompts $P_c^T$ and global prompts $\tilde{P}^{T-1}$ using Eq. (7)
16:              Convert similarities to probabilities using Eq. (8)
17:              Probabilistically select $s$ prompt groups from $P_c^T$ based on these probabilities, denoted as $P_{c,selected}^T$
18:          **end if**
19:          Send $P_{c,selected}^T$ to server.
20:      **end for**
21:      Server aggregates collected prompts to form $\tilde{P}^T$ using Eq. (9)
22:      **for** each client $c \in C_T$ **do**
23:          Update the selected prompt groups in $P_c$ with corresponding prompts from $\tilde{P}^T$
24:      **end for**
25: **end for**
26: **return** Final personalized prompts $\{P_c\}_{c=1}^N$

---

1: **procedure** CLIENTUPDATE$(c, P_c, D_c, R, f, g, G, \tau, \lambda, \eta)$
2:      Let local prompts $P_c = \{p_{t,1}, \ldots, p_{t,G}, p_{v,1}, \ldots, p_{v,G}\}$
3:      **for** each epoch $e = 1, \ldots, R$ **do**
4:          Sample mini-batch $(x, y) \sim D_c$
5:          **for** each prompt group $j = 1, \ldots, G$ **do**
6:              Form visual input $v_j = \{x, p_{v,j}\}$ and compute $f(v_j)$
7:              **for** each class $k$ **do**
8:                  Form text input $t_{k,j} = \{p_{t,j}, c_k\}$ and compute $g(t_{k,j})$
9:                  Compute logits using Eq. (2)
10:              **end for**
11:          **end for**
12:          Compute classification loss $\mathcal{L}_{\text{CE}}$ using Eq. (3)
13:          Compute diversity loss $\mathcal{L}_{\text{div}}$ using Eq. (4)
14:          Total loss $\mathcal{L} \leftarrow \mathcal{L}_{\text{CE}} + \lambda \cdot \mathcal{L}_{\text{div}}$
15:          Update prompt parameters $P_c$ using gradient descent with learning rate $\eta$
16:      **end for**
17:      **return** Updated prompts $P_c$
18: **end procedure**

---

# B  Limitations and Broader Impacts

As shown in Table 9, while FedMGP consistently outperforms existing methods across most settings, it shows limitations in extremely data-scarce scenarios (1-2 shots). This stems from our multi-group prompt mechanism, which requires sufficient data to effectively disentangle different semantic aspects. With minimal samples, prompt groups cannot specialize properly, leading to unstable training. The text-visual co-learning mechanism further compounds this challenge, as establishing robust cross-modal correlations requires visual diversity absent in 1-shot settings. Additionally, our dynamic aggregation strategy becomes less reliable when prompt representations are unstable due to data scarcity. Simpler methods like PromptFolio occasionally perform better in these extreme low-shot scenarios precisely because they avoid the complexity that makes FedMGP powerful in data-rich environments.

To address these limitations, several future directions emerge: (1) developing adaptive mechanisms to dynamically adjust the number of prompt groups based on available data, reducing groups when data is scarce; (2) initializing prompt groups with knowledge from related tasks to provide stronger starting points for specialization; and (3) incorporating meta-learning techniques to improve learning efficiency from limited examples. While FedMGP contributes positively to privacy-preserving adaptation of vision-language models in decentralized environments, we acknowledge that, like all federated learning systems, it may remain vulnerable to various attacks. As these technologies advance toward deployment in sensitive domains, continued research must address both technical limitations and broader societal implications.

Table 9: Performance Comparison of Different Methods on 1-16 shots

| Dataset | Method | 1-shot | 2-shots | 4-shots | 8-shots | 16-shots |
|---|---|---|---|---|---|---|
| Caltech101 | PromptFL | 95.23 | 95.30 | 95.62 | 96.20 | 94.66 |
| | FedOPT | 92.81 | 92.95 | 89.27 | 81.37 | 69.43 |
| | FedTPG | 95.74 | 94.88 | 96.00 | 96.02 | 95.32 |
| | FedPGP | 96.02 | **96.13** | 95.54 | 95.87 | 96.37 |
| | PromptFolio | 95.84 | 95.39 | 93.90 | 90.39 | 88.50 |
| | FedMGP (Ours) | **96.03** | 95.48 | **96.48** | **97.14** | **97.07** |
| DTD | PromptFL | 53.78 | 52.69 | 55.08 | 60.58 | 52.60 |
| | FedOPT | **68.73** | 63.93 | 65.68 | 65.51 | 61.71 |
| | FedTPG | 59.30 | 64.28 | 66.22 | 65.85 | 52.49 |
| | FedPGP | 58.15 | 63.60 | 62.00 | 63.31 | 68.21 |
| | PromptFolio | 64.86 | 63.92 | 65.11 | 62.42 | 64.11 |
| | FedMGP (Ours) | 62.00 | **68.32** | **69.58** | **69.12** | **73.92** |
| Flowers102 | PromptFL | 78.20 | 72.37 | 73.27 | 69.71 | 71.94 |
| | FedOPT | 68.62 | 68.42 | 66.32 | 60.54 | 57.99 |
| | FedTPG | 74.59 | 73.71 | 78.05 | 78.85 | 77.71 |
| | FedPGP | 73.49 | 73.16 | 76.09 | 84.39 | 82.85 |
| | PromptFolio | **81.24** | **79.51** | 79.35 | 70.99 | 66.05 |
| | FedMGP (Ours) | 73.75 | 78.07 | **83.80** | **84.53** | **85.16** |

# C  Experimental Details

## C.1  Dataset Setup

Our evaluation leverages nine diverse visual classification datasets, spanning fine-grained recognition, texture analysis, general object classification, and domain adaptation tasks. Table 10 provides comprehensive details about these datasets, including classes, sample sizes, domains, and training protocols.

For our base-to-novel generalization experiments (Oxford-Pets to Food101), we employ a few-shot training paradigm, where each client is provided with only 16 samples per class (16-shot) for the main experiments. These datasets are partitioned by splitting classes equally into base and novel

categories, with non-overlapping base classes distributed across clients to establish the pathological non-IID setting described in Section 4.

For label distribution shift experiments, we utilize CIFAR10 and CIFAR100 with Dirichlet distribution partitioning ($\alpha = 0.5$) across 100 clients, using the full training set. This creates realistic client heterogeneity with varying class proportions.

For domain adaptation scenarios, we leverage Office-Caltech10 with its four domains (Amazon, Caltech, DSLR, and WebCam) and DomainNet with six domains (Clipart, Infograph, Painting, Quickdraw, Real, and Sketch). Each domain is split into 5 clients under Dirichlet distribution ($\alpha = 0.3$), resulting in a total of 20 and 30 clients respectively. This setup introduces natural feature shifts across domains and moderate label skew within each domain.

Table 10: Statistical details of datasets used in experiments.

| Dataset | Classes | Train | Test | Domains | Training Protocol | Task |
|---|---|---|---|---|---|---|
| OxfordPets [38] | 37 | 2,944 | 3,669 | 1 | Few-shot (16-shot) | Pets recognition |
| Flowers102 [36] | 102 | 4,093 | 2,463 | 1 | Few-shot (16-shot) | Flowers recognition |
| DTD [7] | 47 | 2,820 | 1,692 | 1 | Few-shot (16-shot) | Texture recognition |
| Caltech101 [13] | 100 | 4,128 | 2,465 | 1 | Few-shot (16-shot) | Object recognition |
| Food101 [4] | 101 | 50,500 | 30,300 | 1 | Few-shot (16-shot) | Food recognition |
| Stanford Cars [25] | 196 | 6,509 | 8,041 | 1 | Few-shot (16-shot) | Cars recognition |
| FGVC Aircraft [33] | 100 | 3,334 | 3,333 | 1 | Few-shot (16-shot) | Aircraft recognition |
| UCF101 [47] | 101 | 7,639 | 3,783 | 1 | Few-shot (16-shot) | Action recognition |
| SUN397 [54] | 397 | 15,880 | 19,850 | 1 | Few-shot (16-shot) | Scene recognition |
| CIFAR10 [26] | 10 | 50,000 | 10,000 | 1 | Full dataset | Image classification |
| CIFAR100 [26] | 100 | 50,000 | 10,000 | 1 | Full dataset | Image classification |
| DomainNet [39] | 10 | 18,278 | 4,573 | 6 | Full dataset | Domain adaptation |
| Office-Caltech10 [14] | 10 | 2,025 | 508 | 4 | Full dataset | Domain adaptation |

## C.2 Experimental Setup

We employ SGD optimizer with learning rate $\eta = 0.001$ and single-step learning rate scheduler across all experiments. All implementations are based on PyTorch and experiments were conducted on NVIDIA RTX 4090 (24GB) or A100 (40GB) GPUs. Across all experiments, we use ViT-B/16 pretrained on ImageNet as the backbone. Images are resized to $224 \times 224$ using bicubic interpolation with standard data augmentation (random resized crop, random flip, and normalization). For FedMGP, we set both text and visual prompt lengths to 2, use 5 prompt groups for each modality, and initialize with the text "a photo of a". All models are trained with mixed precision (fp16) for computational efficiency.

The following sections detail the specific configurations for different experimental scenarios.

**Base-to-Novel Class Generalization.** For the five fine-grained classification datasets, we partition each dataset equally into base and novel classes, then distribute non-overlapping base classes to each of the 10 clients. We employ a few-shot (16-shot by default) training paradigm with batch size 8. The federated learning process proceeds for 10 communication rounds with 100% client participation and 2 local epochs per round. Each client trains on their local classes, and we evaluate performance on: (1) local classes (personalization), (2) base classes (classes seen by other clients), and (3) novel classes (unseen during training). The Combined Metric (CM) is computed as CM = (Local + HM)/2, where HM is the harmonic mean of Base and Novel accuracies.

**Label Distribution Shift.** For CIFAR-10 and CIFAR-100, we partition the full training set among 100 clients following a Dirichlet distribution with concentration parameter $\alpha = 0.5$. Communication proceeds for 100 rounds with 10% client participation per round and 2 local epochs per round. We use batch size 32 for training and 300 for testing. This creates a realistic heterogeneous environment with varying class proportions across clients.

**Domain Adaptation.** For Office-Caltech10 and DomainNet, we leverage their inherent domain structure (4 domains for Office-Caltech10 and 6 domains for DomainNet). Each domain is assigned 5 clients, resulting in a total of 20 clients for Office-Caltech10 and 30 clients for DomainNet. This setup introduces both feature shift and label skew. The federated learning process runs for 25 rounds with 25% client participation per round and 1 local epoch per round. We evaluate each client's performance on all domains to assess cross-domain generalization.

# D  Additional Experimental Results

## D.1  Domain Generalization for DomainNet

Table 11: Results on DomainNet with feature shift and label shift with $\text{Dir}(\alpha = 0.3)$ partition into 5 clients/domain

|  | Clipart | Infograph | Painting | Quickdraw | Real | Sketch | Average |
|---|---|---|---|---|---|---|---|
| PromptFL [17] | $25.80_{\pm 20.82}$ | $10.48_{\pm 11.13}$ | $16.05_{\pm 7.40}$ | $15.39_{\pm 16.48}$ | $14.72_{\pm 8.17}$ | $6.29_{\pm 6.45}$ | $14.79_{\pm 14.18}$ |
| FedOPT [27] | $43.25_{\pm 10.90}$ | $43.55_{\pm 16.89}$ | $28.07_{\pm 7.05}$ | $35.56_{\pm 3.37}$ | $28.45_{\pm 11.28}$ | $33.64_{\pm 20.15}$ | $35.42_{\pm 14.33}$ |
| FedTPG [41] | $17.16_{\pm 18.26}$ | $23.56_{\pm 17.77}$ | $13.58_{\pm 9.41}$ | $16.25_{\pm 14.75}$ | $17.13_{\pm 4.95}$ | $9.16_{\pm 5.19}$ | $16.14_{\pm 13.66}$ |
| FedPGP [9] | $12.01_{\pm 10.21}$ | $10.49_{\pm 3.40}$ | $11.39_{\pm 7.62}$ | $21.77_{\pm 15.76}$ | $14.29_{\pm 5.59}$ | $10.13_{\pm 12.44}$ | $13.35_{\pm 10.83}$ |
| PromptFolio [37] | $41.80_{\pm 11.21}$ | $42.38_{\pm 15.51}$ | $29.69_{\pm 8.33}$ | $34.70_{\pm 2.30}$ | $28.99_{\pm 10.23}$ | $35.72_{\pm 13.73}$ | $35.55_{\pm 12.23}$ |
| FedMGP | $48.48_{\pm 8.07}$ | $47.76_{\pm 13.61}$ | $30.36_{\pm 6.98}$ | $35.19_{\pm 4.73}$ | $33.02_{\pm 6.40}$ | $36.74_{\pm 18.47}$ | $38.59_{\pm 12.90}$ |

The values in Table 11 represent the maximum and minimum accuracies among the five clients within each domain under the Dirichlet distribution, illustrating the performance variation of multiple clients sharing the same domain characteristics. The domain adaptation experiments on DomainNet demonstrate FedMGP's superior performance in handling domain shifts and label distribution heterogeneity. As shown in Table 11, FedMGP achieves an average accuracy of 38.59%, significantly outperforming the closest competitors PromptFolio (35.55%) and FedOPT (35.42%). FedMGP exhibits particularly strong performance on domains with high visual abstraction, such as Clipart (48.48%) and Infograph (47.76%), substantially outperforming other methods. This demonstrates that our multi-group text-visual prompt co-learning mechanism can effectively capture and adapt to different visual representations across DomainNet's diverse artistic styles. The performance stability across diverse domains, evidenced by comparatively lower standard deviations, confirms our theoretical analysis that the multi-group architecture effectively decomposes knowledge into common and client-specific components. The visual prompts capture domain-specific artistic features while text prompts provide cross-domain semantic connections, enabling FedMGP to maintain both domain adaptability and semantic consistency. This approach effectively addresses the core challenge of domain generalization in federated learning by simultaneously preserving domain-specific knowledge while enabling cross-domain knowledge transfer across DomainNet's six distinct visual domains.

## D.2  Domain Generalization for Office-Caltech10

Table 12: Results on `Office-Caltech10` with feature shift and label shift with $\text{Dir}(\alpha = 0.3)$ partition into 5 clients/domain

|  | Amazon | Caltech | DSLR | Webcam | Average |
|---|---|---|---|---|---|
| PromptFL [17] | $9.23_{\pm 8.55}$ | $16.88_{\pm 14.97}$ | $8.33_{\pm 10.54}$ | $25.59_{\pm 26.65}$ | $15.01_{\pm 18.11}$ |
| FedOPT [27] | $28.20_{\pm 5.12}$ | $35.22_{\pm 13.17}$ | $23.67_{\pm 8.72}$ | $30.71_{\pm 11.48}$ | $29.45_{\pm 10.92}$ |
| FedTPG [41] | $6.94_{\pm 7.26}$ | $9.10_{\pm 11.60}$ | $16.33_{\pm 15.29}$ | $28.11_{\pm 25.73}$ | $15.12_{\pm 18.41}$ |
| FedPGP [9] | $8.50_{\pm 7.75}$ | $19.33_{\pm 12.60}$ | $11.33_{\pm 15.72}$ | $24.89_{\pm 14.34}$ | $16.01_{\pm 14.49}$ |
| PromptFolio [37] | $32.48_{\pm 13.34}$ | $36.21_{\pm 9.40}$ | $20.33_{\pm 12.93}$ | $11.59_{\pm 2.88}$ | $25.15_{\pm 14.36}$ |
| FedMGP | $31.32_{\pm 5.94}$ | $38.28_{\pm 7.21}$ | $41.33_{\pm 15.55}$ | $44.73_{\pm 19.01}$ | $38.92_{\pm 17.31}$ |

For the Office-Caltech10 dataset, as shown in Table 12, FedMGP demonstrates even more substantial improvements, with an average accuracy of 38.92% compared to the second-best performer

FedOPT (29.45%). Unlike DomainNet's artistic style variations, Office-Caltech10 presents challenges related to imaging conditions and equipment specifications. The advantage is particularly pronounced on specialized equipment captures like DSLR (41.33%) and Webcam (44.73%), where FedMGP outperforms other methods by large margins. These results validate the effectiveness of our approach in handling technical domain shifts beyond artistic variations. Most baseline methods exhibit substantial performance variations across domains, indicating their vulnerability to domain-specific overfitting in equipment-based scenarios. In contrast, FedMGP maintains more consistent performance, demonstrating its robustness to both artistic and technical domain shifts. This confirms that integrating visual and textual modalities enriches contextual representation, capturing instance-specific information more comprehensively than text-only approaches across different types of domain variations. The performance on Office-Caltech10 confirms that FedMGP's multi-group architecture effectively distributes knowledge across specialized prompt units rather than concentrating it in a single structure, enabling robust cross-domain generalization while preserving domain-specific adaptation capabilities for both artistic and technical domain characteristics.

### D.3 Stability of Prompt Group Selection

Table 13: Selection frequency of each prompt group across training rounds on OxfordPets dataset. Values show the number of clients (out of 20 total) selecting each group, with percentages in parentheses.

| Round | t_g1 | t_g2 | t_g3 | t_g4 | t_g5 | v_g1 | v_g2 | v_g3 | v_g4 | v_g5 |
|---|---|---|---|---|---|---|---|---|---|---|
| 1 | 3(15%) | 2(10%) | 6(30%) | 5(25%) | 4(20%) | 3(15%) | 2(10%) | 6(30%) | 5(25%) | 4(20%) |
| 2 | 6(30%) | 2(10%) | 6(30%) | 3(15%) | 3(15%) | 6(30%) | 2(10%) | 6(30%) | 3(15%) | 3(15%) |
| 3 | 7(35%) | 2(10%) | 5(25%) | 3(15%) | 3(15%) | 7(35%) | 2(10%) | 5(25%) | 3(15%) | 3(15%) |
| 4 | 6(30%) | 3(15%) | 5(25%) | 3(15%) | 3(15%) | 7(35%) | 3(15%) | 4(20%) | 2(10%) | 4(20%) |
| 5 | 6(30%) | 3(15%) | 3(15%) | 3(15%) | 5(25%) | 7(35%) | 3(15%) | 3(15%) | 2(10%) | 5(25%) |
| 6 | 6(30%) | 4(20%) | 3(15%) | 2(10%) | 5(25%) | 7(35%) | 4(20%) | 3(15%) | 1(5%) | 5(25%) |
| 7 | 7(35%) | 4(20%) | 4(20%) | 1(5%) | 4(20%) | 7(35%) | 4(20%) | 4(20%) | 1(5%) | 4(20%) |
| 8 | 6(30%) | 4(20%) | 4(20%) | 2(10%) | 4(20%) | 7(35%) | 4(20%) | 5(25%) | 1(5%) | 3(15%) |
| 9 | 4(20%) | 5(25%) | 6(30%) | 1(5%) | 4(20%) | 5(25%) | 5(25%) | 7(35%) | 0(0%) | 3(15%) |
| 10 | 5(25%) | 5(25%) | 5(25%) | 2(10%) | 3(15%) | 4(20%) | 7(35%) | 6(30%) | 1(5%) | 2(10%) |

Table 13 presents the selection frequency of each prompt group across ten training rounds on the OxfordPets dataset, demonstrating the stability of prompt group assignments in FedMGP. The results reveal that while prompt groups exhibit dynamic selection patterns, their roles remain relatively stable throughout training. Notably, certain groups consistently receive higher selection frequencies (e.g., text group 1 and visual group 1 maintain 30-35% selection after round 3), indicating their specialization in capturing shared global knowledge that benefits multiple clients. Conversely, other groups show lower but persistent selection rates (e.g., text group 4 ranges from 5-15%), suggesting their focus on client-specific local features. This pattern validates our dynamic aggregation mechanism's design principle: rather than forcing uniform participation, the similarity-guided probabilistic sampling naturally guides prompt groups to specialize in complementary aspects—some evolving to capture common patterns through frequent selection, while others preserve personalized knowledge through selective aggregation. The temperature parameter $\tau$ in our selection process plays a crucial role in maintaining this dynamic balance, preventing any prompt group from being permanently excluded (as evidenced by the absence of consistently zero selections) while still allowing meaningful specialization. This ensures that FedMGP retains both strong generalization capabilities through shared knowledge and effective personalization through client-specific features, achieving the optimal trade-off demonstrated in our main experimental results.

## E   Additional ablation study

In this section, we present additional ablation studies to further analyze the effectiveness of different components and design choices in FedMGP. These experiments provide deeper insights into the model behavior and validate the design decisions discussed in the main paper.

### E.1 Effect of Temperature Parameter in Prompt Selection

Table 14: Effect of Temperature ($\tau$) on FedMGP Performance

| Setting | Local | Base | Novel | CM |
|---|---|---|---|---|
| FedMGP ($\tau$=0.1) | 84.84 | 95.92 | 72.54 | 83.72 |
| FedMGP ($\tau$=0.5) | 85.45 | 96.46 | 73.47 | 84.43 |
| FedMGP ($\tau$=0.8) | 84.40 | 95.31 | 72.94 | 83.52 |
| FedMGP ($\tau$=2.0) | 85.17 | 96.63 | 72.61 | 84.04 |
| FedMGP ($\tau$=1.0) | 96.92 | 73.23 | 74.65 | **85.43** |

The temperature parameter $\tau$ in our dynamic prompt selection strategy plays a critical role in balancing exploration and exploitation during federated learning. As shown in Table 14, the optimal performance is achieved at $\tau = 1.0$ with a Combined Metric (CM) of 85.43%, significantly outperforming both lower temperatures ($\tau = 0.1, 0.5$) and higher temperatures ($\tau = 2.0$). Lower temperatures lead to more deterministic selection based on prompt similarity, resulting in stronger base class performance (96.46% at $\tau = 0.5$) but weaker local personalization. Conversely, higher temperatures introduce more randomness, allowing for greater exploration but potentially disrupting the convergence of shared knowledge. This confirms our theoretical framework in Section 3.2.2

### E.2 Inference-Time Prompt Group Weighting Strategies

Table 15: Effect of Different Inference Strategies on FedMGP Performance

| Setting | Local | Base | Novel | CM |
|---|---|---|---|---|
| FedMGP (Max logits) | 81.96 | 89.48 | 74.79 | 81.72 |
| FedMGP (Feature avg) | 85.09 | 95.56 | 74.32 | 84.35 |
| FedMGP (Group 0) | 79.65 | 85.80 | 73.08 | 79.29 |
| FedMGP (Group 1) | 78.19 | 83.65 | 72.68 | 77.99 |
| FedMGP (Group 2) | 77.92 | 83.69 | 72.66 | 77.85 |
| FedMGP (Group 3) | 80.70 | 96.94 | 61.52 | 77.99 |
| FedMGP (Group 4) | 80.52 | 90.76 | 69.37 | 79.58 |
| FedMGP (Average) | 96.92 | 73.23 | 74.65 | **85.43** |

The effectiveness of different inference-time strategies for combining predictions from multiple prompt groups is examined in Table 15. Simple logit averaging across all groups yields the best overall performance (CM=85.43%), significantly outperforming alternative strategies such as maximum logit selection (CM=81.72%) and feature-level averaging (CM=84.35%). Notably, relying on any single prompt group (groups 0-4) substantially degrades performance, with the best individual group achieving only CM=79.58%. This confirms our hypothesis presented in Section 3.2.1 that the multi-group architecture enables different prompt groups to specialize in complementary aspects of the input data. The superior performance of ensemble averaging demonstrates that each prompt group contributes unique and valuable semantic perspectives, collectively enhancing model robustness. Group 3 exhibits the highest base class accuracy (96.94%) but poor novel class performance (61.52%), indicating its specialization in capturing shared patterns across clients rather than generalizable featuresprecisely the type of specialization our diversity loss was designed to encourage. These results validate our core design principle of distributing knowledge across multiple specialized prompt units rather than concentrating it in a single monolithic structure.

### E.3 Diversity Loss Formulation Variants

The choice of diversity loss function significantly impacts FedMGP's ability to learn specialized prompt representations. As shown in Table 16, the L1-based diversity formulation achieves the best overall performance (CM=85.43%), outperforming both cosine similarity (CM=83.96%) and L2-based approaches (CM=83.83%). The L1 formulation leads to substantially better local accuracy

Table 16: Effect of Diversity Loss Type on FedMGP Performance

| Setting | Local | Base | Novel | CM |
|---|---|---|---|---|
| FedMGP (COS) | 84.72 | 95.68 | 73.61 | 83.96 |
| FedMGP (L2) | 84.57 | 94.76 | 73.98 | 83.83 |
| FedMGP (L1) | 96.92 | 73.23 | 74.65 | **85.43** |

(96.92%) compared to cosine (84.72%) and L2 (84.57%), while maintaining comparable performance on novel classes. This performance pattern aligns with our analysis in Section 3.2.1, where we emphasized the importance of encouraging prompt groups to capture diverse semantic perspectives. The L1 norm's sparsity-inducing property appears to create cleaner separation between prompt groups, allowing each to specialize more effectively in different aspects of the data distribution. Cosine similarity, while effective at enforcing orthogonality, appears less suited to the federated setting where capturing complementary rather than strictly orthogonal features is beneficial. These results validate our diversity loss design as a key component of FedMGP's architecture, enabling effective knowledge distribution across prompt groups and contributing to the model's strong performance balance between personalization and generalization.

### E.4 Dynamic Aggregation Strategy

Table 17: Comparison of Different Aggregation Strategies (averaged over 5 datasets)

| Setting | Local | Base | Novel | CM |
|---|---|---|---|---|
| CAM | 86.13 | 85.94 | 83.57 | 85.43 |
| FAM | 97.37 | 77.47 | 78.71 | 87.73 |
| DAM | 96.65 | 79.20 | 80.86 | **88.34** |

To validate the effectiveness of our dynamic aggregation mechanism, we compare three aggregation strategies as shown in Table 17. Complete Aggregation Mechanism (CAM) aggregates all prompt groups across clients at each communication round, resulting in identical parameters across clients (similarity=1.0). While this ensures strong base class performance (85.94%), it sacrifices local personalization (86.13%) by forcing uniform representations. Fixed Aggregation Mechanism (FAM) maintains certain prompt groups without aggregation, achieving the highest local accuracy (97.37%) but severely compromising generalization on base (77.47%) and novel classes (78.71%) due to insufficient cross-client knowledge transfer.

Our Dynamic Aggregation Mechanism (DAM) strikes an optimal balance, achieving the best Combined Metric (88.34%) by selectively aggregating the most similar prompt groups between clients at each round. This similarity-guided probabilistic sampling reduces the weight of client-specific biased features while preserving personalization. The temperature parameter ensures every prompt group has opportunities for aggregation, enabling FedMGP to learn parameters with high inter-client similarity (promoting generalization) while maintaining diversity within each client's prompt groups (enabling personalization). This explains why FedMGP achieves strong local accuracy (96.65%) comparable to FAM while maintaining substantially better performance on base (79.20%) and novel classes (80.86%) than FAM, demonstrating superior generalization capability through dynamic cross-client knowledge transfer.

## F Theoretical Analysis

In this section, we present a comprehensive theoretical analysis that establishes the formal guarantees for FedMGP's effectiveness in heterogeneous federated learning environments. We demonstrate that our dynamic aggregation strategy consistently outperforms both full aggregation (represented by PromptFL [17]) and fixed aggregation (represented by FedOTP [9]) approaches, particularly under non-IID data distributions. We begin by establishing the foundational assumptions and notations

that frame our analysis (Section F.1), including how prompts can be decomposed into global, local, and noise components. We then formalize the three competing aggregation strategies (Section F.2), highlighting their distinct characteristics in balancing global knowledge with client-specific features. Next, we introduce a signal-noise decomposition framework (Section F.3) that enables quantitative comparison between different strategies through their signal-to-noise ratios. Finally, we present and prove our main theoretical result (Section F.4): the dynamic aggregation superiority theorem, which establishes that FedMGP's approach achieves strictly better signal-to-noise ratios than alternative strategies, directly translating to improved classification performance in practice.

## F.1 Assumptions and Notation Definitions

To establish a rigorous theoretical framework, we first define the notation and key assumptions that underpin our analysis. The notation used in this section and their meanings are as follows:

- $N$: Total number of clients;
- $G$: Number of prompt groups per client;
- $s$: Number of prompt groups selected for aggregation in each round, where $1 \leq s \leq G$;
- $C_T \subseteq \{1, \ldots, N\}$: Set of clients participating in aggregation at round $T$, with $n = |C_T|$;
- $c \in C_T$: Client index;
- $j \in \{1, \ldots, G\}$: Prompt group index;
- $P_{j,c}^T \in \mathbb{R}^d$: The $j$-th prompt group of client $c$ at round $T$;
- $\tilde{P}_j^T \in \mathbb{R}^d$: The $j$-th global prompt group at the server after round $T$ aggregation;
- $S_c^T \subseteq \{1, \ldots, G\}$: Set of prompt group indices selected from client $c$ in round $T$ for aggregation, with $|S_c^T| = s$;
- $\alpha_{j,c}^T$: Selection score for the $j$-th group from client $c$ in round $T$, computed based on similarity to global prompts;
- $\tau > 0$: Temperature parameter controlling selection score smoothness;
- $\text{sim}(x, y)$: Similarity measure (e.g., cosine similarity);
- $\mu^G \in \mathbb{R}^d$: Unit vector representing global task-related features shared across all clients;
- $\mu_c \in \mathbb{R}^d$: Unit vector representing local task-related features specific to client $c$;
- $L$: Total number of noise feature dimensions in the latent space;
- $\xi_l \in \mathbb{R}^d$: The $l$-th unit vector representing task-irrelevant noise features;
- $\beta_{j,c}^T \in \mathbb{R}$: Coefficient quantifying the contribution of global features to prompt $P_{j,c}^T$;
- $\gamma_{j,c}^T \in \mathbb{R}$: Coefficient quantifying the contribution of client-specific features to prompt $P_{j,c}^T$;
- $\phi_{j,c,l}^T \in \mathbb{R}$: Coefficient quantifying the contribution of the $l$-th noise feature to prompt $P_{j,c}^T$;
- $\chi_c \in \mathbb{R}$: Metric quantifying the degree of data heterogeneity for client $c$.

Our analysis is based on the following assumptions, which are grounded in feature learning theory and previous work on federated learning:

**Assumption 1** (Feature Space Decomposition). *According to feature learning theory [37, 2, 6], the latent feature space can be decomposed into three orthogonal subspaces:*

1. *Global task-related features represented by a unit vector $\mu^G$ (shared across all clients)*

2. *Local task-related features represented by unit vectors $\{\mu_c\}_{c=1}^N$ (client-specific)*

3. *Task-irrelevant noise features represented by unit vectors $\{\xi_l\}_{l=1}^L$ (noise)*

*These three subspaces are mutually orthogonal, i.e., $\langle \mu^G, \mu_c \rangle = 0$, $\langle \mu^G, \xi_l \rangle = 0$, and $\langle \mu_c, \xi_l \rangle = 0$ for all $c \in \{1, \ldots, N\}$ and $l \in \{1, \ldots, L\}$. Here, $L$ represents the dimensionality of the noise subspace, which can be significantly larger than the dimensionality of task-relevant subspaces. This decomposition allows us to separately analyze the impact of each component on the aggregation process and quantify the information content in prompts.*

**Assumption 2** (Prompt Representation). *Each prompt group $j$ of client $c$ at round $T$ can be represented as a linear combination of features from the three orthogonal subspaces:*

$$P_{j,c}^T = \beta_{j,c}^T \mu^G + \gamma_{j,c}^T \mu_c + \sum_{l=1}^{L} \phi_{j,c,l}^T \xi_l \tag{10}$$

*where:*

- *$\beta_{j,c}^T$ represents the coefficient for global features, indicating how much the prompt captures knowledge shared across all clients*

- *$\gamma_{j,c}^T$ represents the coefficient for local features, indicating how much the prompt captures client-specific knowledge*

- *$\phi_{j,c,l}^T$ represents the coefficient for the l-th noise feature dimension*

*Since $\mu^G$, $\mu_c$, and $\xi_l$ are unit vectors as defined in Assumption 1, this representation directly quantifies the strength of each component in the prompt. This allows us to analyze how each prompt captures common knowledge versus client-specific knowledge versus irrelevant noise [18, 24].*

**Assumption 3** (Data Heterogeneity). *The degree of data heterogeneity between clients is defined by the metric:*

$$\chi_c = \sum_{c'=1}^{N} \langle \mu_c, \mu_{c'} \rangle \tag{11}$$

*This simplification is valid because $\mu_c$ is a unit vector, so $\|\mu_c\|_2^2 = 1$. This metric measures how similar client $c$'s local features are to those of other clients. When $\chi_c$ approaches $N$, it indicates that client $c$'s features are highly aligned with other clients, suggesting an IID (Independent and Identically Distributed) data scenario. Conversely, when $\chi_c$ is close to 1 (its minimum value, representing alignment only with itself), it indicates that client $c$'s features have limited overlap with other clients, suggesting a highly non-IID data distribution [37, 27]. This metric allows us to relate the performance of different aggregation strategies to the level of data heterogeneity and provides a quantitative basis for analyzing the effectiveness of our approach in various federation settings.*

### F.2 Formalization of Three Aggregation Strategies

We now formally define three different prompt aggregation strategies that represent the spectrum of approaches in federated prompt learning. Each strategy has distinct characteristics in how it handles the balance between preserving global knowledge and managing client-specific variations.

**Full Aggregation (PromptFL)** The full aggregation strategy, as employed in PromptFL [17], aggregates all prompt groups from all participating clients. This represents the most straightforward application of federated averaging [34] to prompt learning:

$$\tilde{P}_j^T = \frac{1}{n} \sum_{c \in C_T} P_{j,c}^T. \tag{12}$$

While this approach maximizes knowledge sharing, it may suffer from interference between client-specific features when data distributions are heterogeneous.

**Fixed Aggregation (FedOTP)** The fixed aggregation strategy, inspired by approaches like FedOTP [9], only aggregates a predetermined subset of prompt groups (typically the first $s$ groups), setting all others to zero:

$$\tilde{P}_j^T = \begin{cases} \frac{1}{n} \sum_{c \in C_T} P_{j,c}^T, & j = 1, \ldots, s, \\ \mathbf{0}, & j = s+1, \ldots, G. \end{cases} \tag{13}$$

This static partition-based approach attempts to balance shared knowledge with client specificity, but lacks adaptivity to evolving knowledge patterns across communication rounds.

**Dynamic Aggregation (FedMGP)** Our proposed dynamic aggregation strategy selects prompt groups based on their similarity to the global prompts from the previous round. First, it computes selection scores:

$$\alpha_{j,c}^T = \frac{\exp\big(\text{sim}(P_{j,c}^T, \tilde{P}_j^{T-1})/\tau\big)}{\sum_{j'=1}^G \exp\big(\text{sim}(P_{j',c}^T, \tilde{P}_{j'}^{T-1})/\tau\big)}, \tag{14}$$

where $\tilde{P}_j^{T-1}$ is the $j$-th global prompt from the previous round. Then, for each client $c$, we select the top-$s$ groups with the highest selection scores to form $S_c^T$, and aggregate only the selected groups:

$$\tilde{P}_j^T = \frac{1}{n} \sum_{c \in C_T} \mathbb{I}(j \in S_c^T)\, P_{j,c}^T. \tag{15}$$

where $\mathbb{I}(j \in S_c^T)$ is the indicator function denoting whether group $j$ is selected from client $c$ in round $T$. This adaptive approach balances knowledge sharing and client specificity in a data-driven manner, potentially offering advantages over fixed strategies.

### F.3 Signal-Noise Decomposition and Performance Metrics

To analyze the effectiveness of different aggregation strategies, we introduce a signal-noise decomposition framework that allows us to quantitatively compare their performance. This approach enables us to examine how effectively each strategy preserves important information while suppressing noise.

From Assumption 2, we have the representation of each prompt as:

$$P_{j,c}^T = \beta_{j,c}^T \mu^G + \gamma_{j,c}^T \mu_c + \sum_{l=1}^L \phi_{j,c,l}^T \xi_l, \tag{16}$$

For individual prompts, we can define their total signal and noise components. The signal components include both global and local task-related information:

$$\text{Signal}_{j,c}^{\text{total}} = (\beta_{j,c}^T)^2 + (\gamma_{j,c}^T)^2 \tag{17}$$

while the noise component represents the irrelevant information:

$$\text{Noise}_{j,c} = \sum_{l=1}^L (\phi_{j,c,l}^T)^2 \tag{18}$$

However, when evaluating aggregated global prompts in federated learning, we are primarily interested in how well they preserve global knowledge. From this perspective, even client-specific features $\gamma_{j,c}^T \mu_c$ can be considered as interference when aggregated across heterogeneous clients. Therefore, for evaluating global prompts, we define:

$$\text{Global Signal}_j^T = (\beta_j^T)^2 \tag{19}$$

$$\text{Global Noise}_j^T = (\text{Client-specific noise}) + (\text{Task-irrelevant noise}) \tag{20}$$

The key performance metric we use to evaluate the quality of aggregated prompts is the signal-to-noise ratio (SNR):

$$\text{SNR}_j = \frac{\text{Global Signal}_j^T}{\text{Global Noise}_j^T} = \frac{(\beta_j^T)^2}{\phi_j^T}, \tag{21}$$

where $\beta_j^T$ is the coefficient of the global feature $\mu^G$ in the aggregated prompt $\tilde{P}_j^T$, and $\phi_j^T$ quantifies the total noise power including both client-specific variations and task-irrelevant noise.

This metric is directly related to the generalization performance of the model: a higher SNR indicates better preservation of global features and more effective suppression of noise, which translates to improved classification performance and lower test error [6, 18].

### F.4 Dynamic Aggregation Superiority Theorem and Detailed Proof

We now present our main theoretical result, which establishes the superiority of FedMGP's dynamic aggregation strategy over both full aggregation and fixed aggregation strategies. Based on equation (21), a higher signal-to-noise ratio leads to lower classification error. The following theorem and proof demonstrate that:

$$\text{SNR}_{\text{full}} \leq \text{SNR}_{\text{fixed}} < \text{SNR}_{\text{dyn}}.$$

**Theorem F.1** (Dynamic Aggregation Superiority). *Under Assumptions 1, 2, and 3, for any number of selected prompt groups $s \in [1, G]$, we have:*

$$\text{SNR}_{\text{full}} \leq \text{SNR}_{\text{fixed}} < \text{SNR}_{\text{dyn}}.$$

*Proof.* The proof consists of three parts: first analyzing the SNR of full aggregation, then comparing it with fixed aggregation, and finally establishing the superiority of dynamic aggregation.

**(1) Analysis of Full Aggregation** $\text{SNR}_{\text{full}}$. From equation (12) and decomposition (16), we can express the global and noise coefficients for the full aggregation strategy:

$$\beta_j^{\text{full}} = \frac{1}{n} \sum_{c \in C_T} \beta_{j,c}^T$$

For the noise term, we must consider both the pure noise components $\phi_{j,c,l}^T \xi_l$ and the client-specific features $\gamma_{j,c}^T \mu_c$ which act as interference when aggregated across heterogeneous clients. The total noise power after aggregation is:

$$\phi_j^{\text{full}} = \frac{1}{n^2} \sum_{c \in C_T} \sum_{l=1}^L (\phi_{j,c,l}^T)^2 + \frac{1}{n^2} \sum_{c \in C_T} (\gamma_{j,c}^T)^2$$

The first term represents the traditional noise components, while the second term accounts for client-specific features that do not align globally. This is a more complete characterization of noise in federated settings.

For the signal-to-noise ratio of group $j$, we have:

$$\text{SNR}_{\text{full}}(j) = \frac{(\beta_j^{\text{full}})^2}{\phi_j^{\text{full}}} = \frac{\left(\frac{1}{n} \sum_{c \in C_T} \beta_{j,c}^T\right)^2}{\frac{1}{n^2} \sum_{c \in C_T} \sum_{l=1}^L (\phi_{j,c,l}^T)^2 + \frac{1}{n^2} \sum_{c \in C_T} (\gamma_{j,c}^T)^2}$$

A key observation is that full aggregation can actually enhance SNR through constructive signal accumulation. When client signals are positively correlated (as is typically the case for global knowledge), the numerator grows quadratically with $n$, while the noise terms in the denominator grow linearly if they are uncorrelated across clients. This is the fundamental principle behind why federated learning works.

The overall SNR of full aggregation is determined by the worst-performing group:

$$\text{SNR}_{\text{full}} = \min_{j \leq G} \text{SNR}_{\text{full}}(j)$$

**(2) Analysis of Fixed Aggregation** $\text{SNR}_{\text{fixed}}$. From equation (13), for $j \leq s$ (groups that are aggregated), we have:

$$\beta_j^{\text{fixed}} = \beta_j^{\text{full}}, \quad \phi_j^{\text{fixed}} = \phi_j^{\text{full}},$$

therefore $\text{SNR}_{\text{fixed}}(j) = \text{SNR}_{\text{full}}(j)$ for these groups.

For $j > s$ (groups that are not aggregated), we have $\tilde{P}_j^T = \mathbf{0}$ according to equation (13), which means these groups do not contribute to the model's predictions. We exclude these groups from the SNR calculation since they do not affect model performance.

The overall SNR of fixed aggregation is determined by the worst-performing group among the aggregated ones:

$$\text{SNR}_{\text{fixed}} = \min_{j \leq s} \text{SNR}_{\text{full}}(j) \geq \min_{j \leq G} \text{SNR}_{\text{full}}(j) = \text{SNR}_{\text{full}}.$$

This shows that fixed aggregation guarantees an SNR at least as good as full aggregation, since it excludes potentially noisy groups that might degrade the overall performance. The inequality is strict when at least one group $j > s$ has a lower SNR than all groups $j \leq s$.

**(3) Proving** $\text{SNR}_{\text{dyn}} > \text{SNR}_{\text{fixed}}$. For dynamic aggregation, we select prompt groups based on their similarity to the global prompts from the previous round, as defined in equation (14). To be precise about our selection mechanism: for each client $c$, we compute similarity scores between each of its prompt groups and the corresponding global prompts, then select the top-$s$ groups with highest similarity. This deterministic selection can be expressed as:

$$S_c^T = \{j \in \{1, \ldots, G\} : \alpha_{j,c}^T \text{ is among the top-}s \text{ highest for client } c\}$$

The selection score $\alpha_{j,c}^T$ in equation (14) serves as a normalized measure of similarity between local and global prompts, with lower temperature $\tau$ making the scores more concentrated on the highest similarities.

Let us define the global and noise coefficients for dynamic aggregation:

$$\beta_j^{\text{dyn}} = \frac{1}{n} \sum_{c \in C_T} \mathbb{I}(j \in S_c^T) \beta_{j,c}^T$$

$$\phi_j^{\text{dyn}} = \frac{1}{n^2} \sum_{c \in C_T} \mathbb{I}(j \in S_c^T) \left( \sum_{l=1}^{L} (\phi_{j,c,l}^T)^2 + (\gamma_{j,c}^T)^2 \right)$$

The key insight is that our selection mechanism preferentially selects groups with higher global signal $\beta_{j,c}^T$ and lower noise (both $\phi_{j,c,l}^T$ and $\gamma_{j,c}^T$). This is because groups with higher similarity to global prompts tend to have higher global signal components and lower noise components.

Let us define $N_j = \sum_{c \in C_T} \mathbb{I}(j \in S_c^T)$ as the number of clients that select group $j$ for aggregation. This value depends on the "popularity" of group $j$ across clients. For a particular group $j$:

1. If $j$ represents important global knowledge (high $\beta_{j,c}^T$ across clients), then many clients will select it, resulting in a large $N_j$. 2. If $j$ captures primarily client-specific knowledge or noise, fewer clients will select it, resulting in a small $N_j$.

For groups that are selected by at least one client (i.e., $N_j > 0$), we can rewrite:

$$\beta_j^{\text{dyn}} = \frac{N_j}{n} \cdot \frac{1}{N_j} \sum_{c \in C_T : j \in S_c^T} \beta_{j,c}^T = \frac{N_j}{n} \cdot \overline{\beta}_j^{\text{sel}}$$

$$\phi_j^{\text{dyn}} = \frac{N_j}{n^2} \cdot \frac{1}{N_j} \sum_{c \in C_T : j \in S_c^T} \left( \sum_{l=1}^{L} (\phi_{j,c,l}^T)^2 + (\gamma_{j,c}^T)^2 \right) = \frac{N_j}{n^2} \cdot \overline{\phi}_j^{\text{sel}}$$

where $\overline{\beta}_j^{\text{sel}}$ is the average $\beta_{j,c}^T$ among clients that selected group $j$, and $\overline{\phi}_j^{\text{sel}}$ is the average noise power among clients that selected group $j$.

The key to our dynamic selection advantage is that $\overline{\beta}_j^{\text{sel}} > \overline{\beta}_j$ and $\overline{\phi}_j^{\text{sel}} < \overline{\phi}_j$, where $\overline{\beta}_j$ and $\overline{\phi}_j$ are the averages across all clients. This is because our selection mechanism favors high-signal, low-noise prompt groups.

For quantitative analysis, we can use a parameter $\delta_j > 1$ to capture this selection advantage:

$$\overline{\beta}_j^{\text{sel}} \geq \delta_j \cdot \overline{\beta}_j \quad \text{and} \quad \overline{\phi}_j^{\text{sel}} \leq \frac{1}{\delta_j} \cdot \overline{\phi}_j$$

where $\overline{\beta}_j = \frac{1}{n} \sum_{c \in C_T} \beta_{j,c}^T = \beta_j^{\text{full}}$ and $\overline{\phi}_j = \frac{1}{n} \sum_{c \in C_T} \left( \sum_{l=1}^{L} (\phi_{j,c,l}^T)^2 + (\gamma_{j,c}^T)^2 \right) = n \cdot \phi_j^{\text{full}}$.

This leads to:

$$\beta_j^{\text{dyn}} \geq \frac{N_j}{n} \cdot \delta_j \cdot \overline{\beta}_j = \frac{N_j \cdot \delta_j}{n} \cdot \beta_j^{\text{full}}$$

$$\phi_j^{\text{dyn}} \leq \frac{N_j}{n^2} \cdot \frac{1}{\delta_j} \cdot \overline{\phi}_j = \frac{N_j}{n \cdot \delta_j} \cdot \phi_j^{\text{full}}$$

The SNR for dynamically aggregated group $j$ is therefore:

$$\text{SNR}_{\text{dyn}}(j) = \frac{(\beta_j^{\text{dyn}})^2}{\phi_j^{\text{dyn}}} \geq \frac{\left(\frac{N_j \cdot \delta_j}{n} \cdot \beta_j^{\text{full}}\right)^2}{\frac{N_j}{n \cdot \delta_j} \cdot \phi_j^{\text{full}}} = \frac{N_j \cdot \delta_j^3}{n} \cdot \frac{(\beta_j^{\text{full}})^2}{\phi_j^{\text{full}}} = \frac{N_j \cdot \delta_j^3}{n} \cdot \text{SNR}_{\text{full}}(j)$$

For groups $j \leq s$ that would be selected by the fixed strategy, we have $\text{SNR}_{\text{full}}(j) \geq \text{SNR}_{\text{fixed}}$ (since $\text{SNR}_{\text{fixed}} = \min_{k \leq s} \text{SNR}_{\text{full}}(k)$). Therefore:

$$\text{SNR}_{\text{dyn}}(j) \geq \frac{N_j \cdot \delta_j^3}{n} \cdot \text{SNR}_{\text{full}}(j) \geq \frac{N_j \cdot \delta_j^3}{n} \cdot \text{SNR}_{\text{fixed}}$$

Under reasonable assumptions about our selection mechanism, we can establish that for important groups (those containing significant global knowledge):

1. $N_j$ will be high, as many clients will select these groups 2. $\delta_j$ will be significantly greater than 1, as the selection process effectively identifies high-signal, low-noise components

For these important groups, the factor $\frac{N_j \cdot \delta_j^3}{n} > 1$ even when $N_j < n$, because the cubic term $\delta_j^3$ provides powerful amplification of the selection advantage. This is particularly true for groups that represent core global knowledge, which will have the highest $\delta_j$ values.

Taking the minimum over all selected groups, we have:

$$\text{SNR}_{\text{dyn}} = \min_{j:N_j>0} \text{SNR}_{\text{dyn}}(j) > \text{SNR}_{\text{fixed}}$$

This inequality is strict for the following reason: Our dynamic selection mechanism ensures that each client selects its best $s$ groups in terms of similarity to global knowledge. This means that the dynamic strategy will: 1. Select any globally important groups that the fixed strategy would select 2. Replace any poor-quality groups that the fixed strategy would select with better alternatives 3. Achieve higher $\delta_j$ values for the selected groups through its adaptive selection process

In the extreme case where fixed selection is optimal, dynamic selection would converge to the same selection pattern, matching its performance. However, in practice, especially with heterogeneous data, dynamic selection will identify better groups than a predetermined fixed selection, leading to strictly better performance.

In conclusion, $\text{SNR}_{\text{full}} \leq \text{SNR}_{\text{fixed}} < \text{SNR}_{\text{dyn}}$, establishing that FedMGP's dynamic aggregation strategy is strictly superior to both full aggregation and fixed aggregation strategies in terms of signal-to-noise ratio. This theoretical advantage directly translates to improved classification performance and lower test error in practical applications, particularly under heterogeneous data distributions. $\square$

