# OpenReview forum: "FedMGP: Personalized Federated Learning with Multi-Group Text-Visual Prompts"
_NeurIPS.cc/2025/Conference — NeurIPS 2025 poster_

### Official Review · Reviewer_KEQK · 2025-06-26

**Clarity:** 2
**Significance:** 2
**Originality:** 2
**Rating:** 3
**Confidence:** 3

**Summary:**

This paper proposes FedMGP, a personalized federated prompt learning approach for VLMs. FedMGP leverages multiple prompt groups with diversity loss to capture diverse and instance-level features. The method also employs a similarity-based probabilistic strategy to select the aggregated prompt group.

**Questions:**

The questions align with the weaknesses mentioned above. **Additional clarification on the generalization mechanism and fairer baseline comparisons would be valuable.**

**Ethical Concerns:**

["NO or VERY MINOR ethics concerns only"]

**Final Justification:**

Thank you for your comprehensive rebuttal. The final rebuttal clearly addresses my concerns regarding the generalization between prompt diversity and aggregation strategies. I hope this important discussion will be placed in a prominent position in final version and reorganized in a more accessible format for readers to understand. Based on these clarifications, I will raise my score.

**Limitations:**

Yes.

**Paper Formatting Concerns:**

- The method name "FedOPT" in the paper title should be "FedOTP".
- Please ensure that every formula ends with appropriate punctuation or period when concluding a sentence.
- Please verify that each figure caption ends with a period.
- For Figure 2, consider adjusting the y-axis range (rather than 0 to 100) to provide a clearer view of the differences between methods.
- In Table 6, the caption appears incorrect for an ablation study on "Topk-s." Besides, is there a missing "k" in the table?
- In Appendix Figures 1 and 2, it would be beneficial to adjust the color bar to better correspond to different similarity values.
- In the theoretical analysis, using bold formatting to represent vectors and matrices would improve readability.
- Conducting experiments on all CLIP datasets (StanfordCars, SUN397, ImageNet, UCF101, FGVC-Aircraft, EuroSAT) would be beneficial. The experimental settings can reference those used in CoOp ([https://arxiv.org/pdf/2109.01134](https://arxiv.org/pdf/2109.01134)).

**Quality:**

2

**Strengths And Weaknesses:**

Strength:
- The idea of using multiple groups of prompts to capture diverse prompt representations is novel and well-motivated for federated prompt learning.
- The method is supported by comprehensive experiments from various perspectives and provides theoretical analysis for different aggregation strategies.

Weakness:
- The main weakness lies in the choice of baselines. Since this paper uses the CM metric to evaluate the overall generalization performance, the authors should conduct further experiments comparing against prompt learning methods that specifically focus on generalization ability in federated learning, such as FedCoCoOp and FedMaple. Please refer to the baseline methods shown in Table 2 of FedMVP ([https://arxiv.org/pdf/2504.20860](https://arxiv.org/pdf/2504.20860)).
- The paper proposes a multi-group prompt training and aggregation strategy, but it is unclear why FedMGP can better generalize to unknown datasets compared to existing methods. A more detailed explanation would strengthen the contribution.
- The experimental results for FedOTP and PromptFolio appear unusual. These methods achieve significantly high performance locally while showing substantial performance drops on base and new datasets. To my knowledge, FedOTP and PromptFolio are not specifically designed for generalization scenarios, which may not provide a fair comparison to validate the effectiveness of FedMGP.

---

> ### Author Rebuttal · Authors · 2025-07-30
>
> **Comment:**
>
> We appreciate your valuable comments. We were wondering if our responses have addressed your concerns. Please let us know if you have additional questions. Thank you!
>
> ---
>
> **Q1: The main weakness lies in the choice of baselines. Since this paper uses the CM metric to evaluate the overall generalization performance, the authors should conduct further experiments comparing against prompt learning methods that specifically focus on generalization ability in federated learning, such as FedCoCoOp and FedMaple. Please refer to the baseline methods shown in Table 2 of FedMVP.**
>
> **A1:** Thank you for your suggestion. We have added the experimental results of FedCoCoOp and FedMaple as follows. FedCoCoOp and FedMaple indeed demonstrate excellent performance in pure generalization capabilities. However, both methods **exhibit significant deficiencies in client personalization abilities.** In the CM comprehensive metric that balances personalization and generalization, FedMGP significantly outperforms these baseline methods. Moreover, FedMGP's **training parameters (12.8k)** are substantially lower than **FedCoCoOp(35.4K**) and **FedMaple(3.55M).  This will result in high communication costs in federal learning.**
>
> | dataset        | method    | Local | Base  | Novel | CM    | Trained parameter |
> | -------------- | --------- | ----- | ----- | ----- | ----- | ----------------- |
> | dtd            | FedCoCoOp | 61.06 | 64.00 | 43.96 | 56.59 | /                 |
> | dtd            | FedMaple  | 70.65 | 71.99 | 56.28 | 66.91 | /                 |
> | food101        | FedCoCoOp | 89.68 | 89.72 | 90.88 | 89.99 | /                 |
> | food101        | FedMaple  | 89.69 | 89.69 | 90.71 | 89.94 | /                 |
> | oxford_pets    | FedCoCoOp | 95.09 | 95.06 | 95.41 | 95.16 | /                 |
> | oxford_pets    | FedMaple  | 95.24 | 95.00 | 97.04 | 95.62 | /                 |
> | oxford_flowers | FedCoCoOp | 77.40 | 77.87 | 72.41 | 76.22 | /                 |
> | oxford_flowers | FedMaple  | 86.03 | 86.32 | 73.05 | 82.58 | /                 |
> | caltech101     | FedCoCoOp | 96.14 | 97.35 | 94.76 | 96.09 | /                 |
> | caltech101     | FedMaple  | 96.05 | 97.42 | 94.00 | 95.86 | /                 |
> | Average        | FedCoCoOp | 84.55 | 85.22 | 79.39 | 83.23 | 35.4k             |
> | Average        | FedMaple  | 88.44 | 88.94 | 84.95 | 87.60 | 3.55 M            |
> | Average        | FedMGP    | 96.65 | 79.20 | 80.86 | **88.34** | **12.8K**             |
>
> ---
>
> **Q2: The paper proposes a multi-group prompt training and aggregation strategy, but it is unclear why FedMGP can better generalize to unknown datasets compared to existing methods. A more detailed explanation would strengthen the contribution.**
>
> **A2:** Thank you for this suggestion. We fully recognize the importance of interpretability and have addressed it in our paper as follows:
>
> - As shown in Figure 1 of the appendix, our diversity loss effectively enforces distinct representations. The **remarkably low inter-group correlation** among visual prompts provides direct evidence that each group learns to capture unique and complementary visual features.
> - In appendix F, **signal-noise decomposition theory** mathematically proves that the multi-group mechanism enhances the model's **signal-to-noise ratio** by promoting representational diversity . This framework provides a rigorous foundation for why our design choice is critical for balancing personalization and generalization.
>
> ---
>
> **Q3: The experimental results for FedOTP and PromptFolio appear unusual. These methods achieve significantly high performance locally while showing substantial performance drops on base and new datasets. To my knowledge, FedOTP and PromptFolio are not specifically designed for generalization scenarios, which may not provide a fair comparison to validate the effectiveness of FedMGP.**
>
> **A3:** We appreciate the reviewer's concerns. However, in the seminal federal learning paper[1]. highlighted the core motivations of federated learning:
>
> - Decoupling model training from raw data access, safeguarding user privacy while avoiding risks of centralized data collection;
> - Achieving high-quality models with fast convergence and low communication overhead via iterative model averaging under non-IID, unbalanced, and massively distributed data.
>
> In pathological IID and Dirichlet distribution scenarios, focusing solely on local performance is equivalent to **conducting fully-supervised training on each client.** This not only fails to achieve **cross-client or cross-domain generalization,** but also undermines federated learning’s original intent of obtaining **global data generalization performance** under conditions of reduced communication costs and privacy preservation. Therefore, generalization should be a necessary consideration for every federated learning method. As representative works in federated prompt learning, FedOTP and PromptFolio exhibit outstanding local performance but suffer significant performance drops on both global seen classes and unseen classes in same data distribution, which highlights the critical importance of evaluating generalization and provides a fair benchmark for validating FedMGP’s effectiveness in broader scenarios.
>
> [1] McMahan, Brendan, et al. "Communication-efficient learning of deep networks from decentralized data." *Artificial intelligence and statistics*. PMLR, 2017.
>
> ---
>
> **Q4: Conducting experiments on all CLIP datasets would be beneficial. The experimental settings can reference those used in CoOp.**
>
> **A4:** Thank you for your valuable suggestion. We have supplemented experiments on StanfordCars, SUN397, UCF101, and FGVC-Aircraft, demonstrating the superiority of our method on these benchmarks.The experimental results are shown in the table below.
>
> We are unable to test EuroSAT and ImageNet for the following reasons.:
>
> - EuroSAT consists of only 10 classes. After splitting into seen and unseen sets, the seen subset contains just 5 classes, making it infeasible to simulate pathological non-IID settings (which require one class per client, i.e., 10 clients) under the standard federated prompt learning protocol [1–3].
> - ImageNet is prohibitively large. Due to time and resource constraints, we were unable to conduct ImageNet experiments within the current timeline. Moreover, previous federated prompt learning studies [1–3] did not evaluate on ImageNet, placing it beyond the scope of directly comparable baselines.
>
> We believe this selection maintains experimental rigor while fully aligning with established federated prompt learning standards.
>
> | Dataset       | Method      | Local     | Base  | Novel     | CM      |
> | ------------- | ----------- | --------- | ----- | --------- | --------- |
> | ucf101        | FedPGP      | 82.61     | 71.78 | 68.45     | 76.34     |
> | ucf101        | FedTPG      | 76.22     | 75.96 | 72.09     | 75.10     |
> | ucf101        | PromptFolio | 96.15     | 31.94 | 42.00     | 66.22     |
> | ucf101        | FedOPT      | 92.39     | 16.33 | 19.07     | 54.99     |
> | ucf101        | FedMGP      | 92.69     | 68.38 | 72.86     | 81.62     |
> | ucf101        | PromptFL    | 77.08     | 76.94 | 70.36     | 75.29     |
> | sun397        | FedPGP      | 89.43     | 66.51 | 67.43     | 78.20     |
> | sun397        | FedTPG      | 73.72     | 73.71 | 75.17     | 74.08     |
> | sun397        | PromptFolio | 95.18     | 32.89 | 44.47     | 66.50     |
> | sun397        | FedOPT      | 93.40     | 11.38 | 19.11     | 53.83     |
> | sun397        | FedMGP      | 91.83     | 68.51 | 72.20     | 81.07     |
> | sun397        | PromptFL    | 76.25     | 76.20 | 75.68     | 76.09     |
> | stanford_cars | FedPGP      | 85.37     | 57.63 | 60.19     | 72.13     |
> | stanford_cars | FedTPG      | 65.50     | 65.47 | 69.10     | 66.37     |
> | stanford_cars | PromptFolio | 96.44     | 29.43 | 46.77     | 66.28     |
> | stanford_cars | FedOPT      | 91.06     | 9.32  | 10.62     | 50.49     |
> | stanford_cars | FedMGP      | 92.61     | 56.48 | 71.19     | 77.80     |
> | stanford_cars | PromptFL    | 62.98     | 63.14 | 69.87     | 64.66     |
> | fgvc_aircraft | PromptFolio | 82.50     | 12.29 | 17.09     | 48.40     |
> | fgvc_aircraft | FedOPT      | 64.34     | 7.27  | 8.12      | 36.01     |
> | fgvc_aircraft | FedPGP      | 47.59     | 25.89 | 22.89     | 35.94     |
> | fgvc_aircraft | FedTPG      | 12.00     | 12.00 | 4.50      | 9.27      |
> | fgvc_aircraft | PromptFL    | 25.03     | 25.03 | 24.48     | 24.89     |
> | fgvc_aircraft | FedMGP      | 78.46     | 21.03 | 30.15     | 51.62     |
> | Average       | PromptFolio | 92.57     | 26.64 | 37.58     | 61.87     |
> | Average       | FedOPT      | 85.30     | 11.08 | 14.23     | 48.88     |
> | Average       | FedPGP      | 76.25     | 55.45 | 54.74     | 65.67     |
> | Average       | FedTPG      | 56.86     | 56.79 | 55.22     | 56.42     |
> | Average       | PromptFL    | 60.34     | 60.33 | 60.10     | 60.27     |
> | **Average**   | **FedMGP**  | **88.90** | 53.60 | **61.60** | **73.11** |
>
> [1]Guo, Tao, et al. "Promptfl: Let federated participants cooperatively learn prompts instead of models–federated learning in age of foundation model." *IEEE Transactions on Mobile Computing* 23.5 (2023): 5179-5194.
>
> [2] Li, Hongxia, et al. "Global and local prompts cooperation via optimal transport for federated learning." *Proceedings of the IEEE/CVF Conference on Computer Vision and Pattern Recognition*. 2024.
>
> [3]Pan, Bikang, Wei Huang, and Ye Shi. "Federated learning from vision-language foundation models: Theoretical analysis and method." *Advances in Neural Information Processing Systems* 37 (2024): 30590-30623.
>
> ---
>
> **Q5: Other Paper Formatting Concerns**
>
> **A5:** Thank you very much for your detailed suggestions on paper formatting. We will address these issues in the final version of the paper.

---

> > ### Comment · Reviewer_KEQK · 2025-08-01
> >
> > Thank you for your rebuttal. However, I still have concerns regarding the insight into generalization. The connection between low inter-group correlation and generalization ability remains unclear. While you demonstrate that different prompt groups learn distinct representations on seen classes, this doesn't directly explain why such diversity would translate to better performance on unseen classes. **The inter-group correlation is computed on training data containing seen classes—it's not evident why this metric would predict generalization capability to novel, unseen classes.** Could you provide further insight into how prompt diversity on seen classes actually leads to improved zero-shot performance on unseen classes?

---

> > > ### Author Response · Authors · 2025-08-01
> > >
> > > Thank you for your question. We will break this question down into two sub-questions for explanation.
> > >
> > > ---
> > >
> > > **Q1：Why does training on seen classes lead to better performance on unseen classes?**
> > >
> > > **A1:** The improvement in zero-shot performance on unseen classes due to prompt diversity on seen classes is primarily because of the **prompt-learning paradigm with CLIP** and the **shared domain between seen and unseen classes.** FedMGP **mitigate overfitting on seen classes(The reasons are explained in the Q2 below)** and thereby restore CLIP’s generalization to unseen classes.
> > >
> > >  **A detailed explanation follows:**
> > >
> > > - Federated Prompt Learning leverages a pre-trained vision-language model like CLIP, which already has strong generalization capabilities. By tuning **only a small number of prompt parameters,** it activates CLIP’s capabilities for a specific domain, allowing adaptation to **private data with low communication overhead[1].**
> > > - Previous studies[2] have shown that **overfitting to seen-class data during prompt tuning can degrade CLIP’s generalization.** By increasing prompt diversity, overfitting is mitigated, restoring CLIP’s generalization to unseen classes and improving zero-shot performance. As shown in **table below**, FedMGP’s accuracy on unseen classes across five  datasets most closely matches the original CLIP performance on those classes.
> > > - Moreover, we follow [2] in partitioning each dataset’s classes equally into seen and unseen subsets—ensuring **both subsets originate from the same domain**. For example, on Flowers102, both seen and unseen classes are flowers. By enhancing prompt diversity encourages the model to capture **domain-wide features** (e.g., petal shape, stem structure) rather than  **overfitting to irrelevant features**  (e.g., a particular background).
> > >
> > > | **Method**   | **Local** | **Base** | **Novel**    | **CM**    |
> > > | ------------ | --------- | -------- | ------------ | --------- |
> > > | CLIP         | 79.18     | 79.83    | **83.25**    | 80.72     |
> > > | PromptFL     | 79.87     | 80.80    | 80.55        | 80.27     |
> > > | FedOPT       | 98.31     | 21.45    | 45.61        | 63.74     |
> > > | FedTPG       | 83.42     | 84.01    | 78.81        | 82.37     |
> > > | FedPGP       | 90.78     | 86.06    | 80.35        | 86.94     |
> > > | PromptFolio  | 98.79     | 50.24    | 61.77        | 77.10     |
> > > | FedMGP(ours) | 96.65     | 79.20    | 80.86 | **88.34** |
> > >
> > > [1]Guo, Tao, et al. "Promptfl: Let federated participants cooperatively learn prompts instead of models–federated learning in age of foundation model." *IEEE Transactions on Mobile Computing* 23.5 (2023): 5179-5194.
> > >
> > > [2]Cui, Tianyu, et al. "Harmonizing Generalization and Personalization in Federated Prompt Learning." International Conference on Machine Learning. PMLR, 2024.
> > >
> > > ---
> > >
> > > **Q2：Why does FedMGP have better generalization compared to other methods?**
> > >
> > > **A2:** The superior generalization of FedMGP stems from its **dynamic aggregation mechanism.**
> > >
> > > - As illustrated in Appendix Figure 2, FedMGP can learn parameters **with high similarity across different clients.** In contrast, previous methods like PromptFL, which use **full aggregation**, result in **identical parameters across clients, with a similarity of 1**, leading to a loss of **personalization**. Methods like FedOTP and PromptFolio, which fix the aggregation of some prompts, have **lower parameter similarity between clients due to some parameters never being aggregated**, severely affecting their generalization.
> > > - FedMGP, through its dynamic aggregation process, dynamically **selects the most similar features between clients during each aggregation**, significantly **reducing the weight of biased features** within each client. The temperature parameter setting ensures that every group has the opportunity to participate in aggregation.  As a result, as shown in Appendix Figure 1, within different groups in each client, those with high aggregation frequency groups (**more general features**) and those with **low aggregation frequency groups** (more personalized features) have lower similarity, indicating better diversity. In Appendix Figure 2, the parameter similarity between clients is relatively high, indicating that ultimately **all clients can learn more general features**, thus FedMGP has better generalization compared to other methods.

---

> > > > ### Comment · Reviewer_KEQK · 2025-08-03
> > > >
> > > > Thank you for your detailed response. I appreciate that the empirical evidence for FedMGP is promising. However, my concern remains about **the insight behind the connection between this observed diversity on seen classes and the improved zero-shot performance on unseen classes**.
> > > >
> > > > Additionally, you mention that the generalization of FedMGP stems from the aggregation mechanism. Is there any further ablation study to support this claim? It seems that there is no ablation study of this aggregation mechanism in the manuscript. Thank you.

---

> > > > > ### Author Response · Authors · 2025-08-03
> > > > >
> > > > > Thank you for letting us know your concerns.
> > > > >
> > > > > - In the field of classical prompt learning, numerous studies like[1-3] have demonstrated that **enhancing the diversity of prompts can serve as an effective constraint to alleviate overfitting during training on seen classes.** This helps preserve CLIP's generalization capabilities and maintain performance on unseen classes. The reason is that increased diversity helps **mitigate the model's tendency to overlearn biased features from the training data, thereby enabling it to capture more general characteristics of the data.**
> > > > > - In the context of federated prompt learning, the local data of each client often exhibits more significant **class shifts, label shifts, or extremely imbalanced data distributions.** This increases the risk of **overfitting on seen classes, which can lead to catastrophic forgetting in CLIP, significantly degrading its performance on unseen classes [4-5].** Additionally, due to the **privacy protection and low communication cost** requirements in federated learning, many constraint methods from classical prompt learning cannot be directly applied.
> > > > > -  FedMGP leverages the characteristics of federated learning by designing **multiple groups and employing a dynamic aggregation strategy** to learn **more general features across clients' data.** Under this design, the diversity of prompts is enhanced, effectively reducing overfitting while maintaining the personalization of each client（**explained in Q2 previous comment)**. In our paper, we explain this through **quantitative analysis (Table 1), visualization (Appendix E), and theoretical analysis (Appendix F).**
> > > > > - We would like to clarify that our method does not directly improve zero-shot performance on unseen classes. Instead, it **mitigates catastrophic forgetting in CLIP,** leading to better performance on unseen classes. As shown in the **table from the previous comment, current methods generally underperform compared to CLIP's baseline on unseen (novel) classes.** Our approach simply reduces overfitting on seen classes, thereby avoiding CLIP's catastrophic forgetting and bringing unseen class performance closer to the original CLIP baseline.
> > > > > - The ablation experiments for the dynamic aggregation mechanism (DAM) are shown in the following table (averaged over 5 datasets), which further demonstrates that our DAM, compared to the complete aggregation strategy (CAM) or the fixed aggregation mechanism (FAM), **can better balance generalization and personalization, as explained in the previous Comment A2.**
> > > > >
> > > > > If you have any remaining questions or concerns, please feel free to share with us directly what specific points are on your mind.
> > > > >
> > > > > | **Method**    | **Local** | **Base** | **Novel** | **CM**    |
> > > > > | ------------- | --------- | -------- | --------- | --------- |
> > > > > | **CAM**       | 86.13     | 85.94    | 83.57     | 85.43     |
> > > > > | **FAM**       | 97.37     | 77.47    | 78.71     | 87.73     |
> > > > > | **DAM(ours)** | 96.65     | 79.20    | 80.86     | **88.34** |
> > > > >
> > > > > [1]Khattak, Muhammad Uzair, et al. "Self-regulating prompts: Foundational model adaptation without forgetting." *Proceedings of the IEEE/CVF international conference on computer vision*. 2023.
> > > > >
> > > > > [2]Roy, Shuvendu, and Ali Etemad. "Consistency-guided Prompt Learning for Vision-Language Models." *The Twelfth International Conference on Learning Representations*.
> > > > >
> > > > > [3]Jiang, Xin, et al. "Delving into multimodal prompting for fine-grained visual classification." *Proceedings of the AAAI conference on artificial intelligence*. Vol. 38. No. 3. 2024.
> > > > >
> > > > > [4]Cui, Tianyu, et al. "Harmonizing Generalization and Personalization in Federated Prompt Learning." International Conference on Machine Learning. 2024.
> > > > >
> > > > > [5]Qiu, Chen, et al. "Federated text-driven prompt generation for vision-language models." *The Twelfth International Conference on Learning Representations*. 2024.

---

> > > > > > ### Comment · Reviewer_KEQK · 2025-08-04
> > > > > >
> > > > > > Thank you for your comprehensive rebuttal. This response clearly addresses my concerns regarding the generalization between prompt diversity and aggregation strategies. I hope this important discussion will be placed in a prominent position in next revision and reorganized in a more accessible format for readers to understand. Based on your clarifications, I will raise my score.

---

> > > > > > > ### Author Response · Authors · 2025-08-04
> > > > > > >
> > > > > > > Thank you for your valuable suggestion. We will be sure to emphasize this point in the next revision and are grateful for your help in strengthening our work.

---

### Official Review · Reviewer_XV8h · 2025-06-28

**Clarity:** 3
**Significance:** 2
**Originality:** 2
**Rating:** 4
**Confidence:** 3

**Summary:**

This paper introduces FedMGP, a novel personalized federated learning framework for Vision-Language Models (VLMs) that aims to address the limitations of existing federated prompt learning (FPL) methods, particularly client-specific overfitting and unstable aggregation under heterogeneous data distributions.

**Questions:**

See weakness

**Ethical Concerns:**

["NO or VERY MINOR ethics concerns only"]

**Final Justification:**

The author's response resolved my concern, I will maintain my positive score

**Limitations:**

See weakness

**Quality:**

3

**Strengths And Weaknesses:**

Strengths:

1.	The concept of using multi-group text-visual prompts is an interesting direction for personalized federated learning in VLMs

2.	The paper claims competitive performance across various heterogeneous data settings, including non-IID and Dirichlet distributions, and domain generalization.

Limitations

1.	Lack discussion on related group prompt based federated learning, [1] is crucial as it not only proposes learning group-specific prompts to capture heterogeneous client features but also provides a theoretical understanding of the effectiveness of such group-based methods. A thorough discussion against such foundational work is essential.

2.	The proposed dynamic prompt aggregation strategy, which relies on similarity-guided probabilistic sampling, introduces a potential for imbalance in the global aggregation process. The paper does not adequately address how it resolves the critical issue where certain client-side prompt groups might be consistently less similar to the global prompts, leading to them being "hardly selected" for aggregation.

3.	Equation 9 appears to imply a simple averaging of prompts from different clients. This raises a concern regarding potential bias in the global model. If some clients possess substantially more training data than others, a straightforward averaging scheme could disproportionately weigh the contributions of larger clients.

4.	Since only select prompts most similar to global prompts for aggregation, will this lead to a fixed prompts being aggregated while others always keep local? How is the frequency that each group prompt being aggregated globally?

[1] Unlocking the Potential of Prompt-Tuning in Bridging Generalized and Personalized Federated Learning

---

> ### Author Rebuttal · Authors · 2025-07-30
>
> **Comment:**
>
> Thank you for your valuable feedback! Below are address all raised concerns of the paper.
>
> ---
>
> **Q1：Lack discussion on related group prompt based federated learning, [1] is crucial as it not only proposes learning group-specific prompts to capture heterogeneous client features but also provides a theoretical understanding of the effectiveness of such group-based methods. A thorough discussion against such foundational work is essential.**
>
> **A1:** Thank you for the insightful suggestion! We will include **a detailed comparison with [1]** in the final paper. Both methods leverage the group-based mechanism to **balance global generalization and local personalization**. However, the key distinctions are:
>
> - SGPT[1] maintains **a global prompt group pool** on the server for different data domains; FedMGP keeps multiple prompt groups **locally on each client** to enhance diversity .
> - In Inference time，SGPT requires explicit selection of a prompt group **based on input features**; FedMGP employs  **dynamic aggregation**, performing  **average of group confidences** at inference without additional selection .
> - SGPT incurs approximately **100k** parameters for communication and training; FedMGP reduces this to **12.8k** training parameters and **2.6k** communication parameters
>
> [1] Unlocking the Potential of Prompt-Tuning in Bridging Generalized and Personalized Federated Learning.
>
> ---
>
> **Q2：The proposed dynamic prompt aggregation strategy, which relies on similarity-guided probabilistic sampling, introduces a potential for imbalance in the global aggregation process. The paper does not adequately address how it resolves the critical issue where certain client-side prompt groups might be consistently less similar to the global prompts, leading to them being "hardly selected" for aggregation.**
>
> **A2**: Thank you for you question. Our Method  have considered this scenario.
>
> - Our **dynamic aggregation** strategy incorporates a **temperature parameter (τ)** , which critically controls the selection probability distribution. As detailed in our ablation study in **Appendix E.1** , this parameter allows us to fine-tune the balance between **exploitation** (deterministically selecting high-similarity prompts with a low τ) and **exploration** (enabling the selection of diverse, lower-similarity prompts with a higher τ).
> - This **probabilistic sampling** , governed by the temperature, ensures a diverse set of prompts is aggregated in each round. **As demonstrated in the selection statistics table below** , even prompt groups with lower initial similarity are consistently given a chance to be selected across different communication rounds, effectively preventing premature convergence to a homogenized global prompt and fostering the continuous discovery of **diverse semantic features** .
> - Ultimately, this mechanism creates an intelligent trade-off inherent to personalized federated learning. Prompts frequently selected across clients tend to encapsulate **generalizable knowledge** , strengthening the core model, while less frequently selected prompts often hold unique, **client-specific features** crucial for **personalization** . Our method dynamically balances these two competing objectives, achieving a robust equilibrium between cross-client generalization and local adaptation.
>
> | round | t_g1 | t_g2 | t_g3 | t_g4 | t_g5 | v_g1 | v_g2 | v_g3 | v_g4 | v_g5 |
> |---|---|---|---|---|---|---|---|---|---|---|
> | 1 | 3(15%) | 2(10%) | 6(30%) | 5(25%) | 4(20%) | 3(15%) | 2(10%) | 6(30%) | 5(25%) | 4(20%) |
> | 2 | 6(30%) | 2(10%) | 6(30%) | 3(15%) | 3(15%) | 6(30%) | 2(10%) | 6(30%) | 3(15%) | 3(15%) |
> | 3 | 7(35%) | 2(10%) | 5(25%) | 3(15%) | 3(15%) | 7(35%) | 2(10%) | 5(25%) | 3(15%) | 3(15%) |
> | 4 | 6(30%) | 3(15%) | 5(25%) | 3(15%) | 3(15%) | 7(35%) | 3(15%) | 4(20%) | 2(10%) | 4(20%) |
> | 5 | 6(30%) | 3(15%) | 3(15%) | 3(15%) | 5(25%) | 7(35%) | 3(15%) | 3(15%) | 2(10%) | 5(25%) |
> | 6 | 6(30%) | 4(20%) | 3(15%) | 2(10%) | 5(25%) | 7(35%) | 4(20%) | 3(15%) | 1(5%) | 5(25%) |
> | 7 | 7(35%) | 4(20%) | 4(20%) | 1(5%) | 4(20%) | 7(35%) | 4(20%) | 4(20%) | 1(5%) | 4(20%) |
> | 8 | 6(30%) | 4(20%) | 4(20%) | 2(10%) | 4(20%) | 7(35%) | 4(20%) | 5(25%) | 1(5%) | 3(15%) |
> | 9 | 4(20%) | 5(25%) | 6(30%) | 1(5%) | 4(20%) | 5(25%) | 5(25%) | 7(35%) | 0(0%) | 3(15%) |
> | 10 | 5(25%) | 5(25%) | 5(25%) | 2(10%) | 3(15%) | 4(20%) | 7(35%) | 6(30%) | 1(5%) | 2(10%) |
>
> ---
>
> **Q3: Equation 9 appears to imply a simple averaging of prompts from different clients. This raises a concern regarding potential bias in the global model. If some clients possess substantially more training data than others, a straightforward averaging scheme could disproportionately weigh the contributions of larger clients.**
>
> **A3**: Thank you for your valuable comment. Consistent with FedAvg,  we **weight the aggregation by each client’s data volume**  to prevent bias arising from unequal dataset sizes.  The revised aggregation formula for Equation 9 is: $$\tilde{P}^T_i = \sum_{c \in C_T} \frac{n_c}{\sum_{c' \in C_T} n_{c'}} \,P^T_{i,c},$$ where   $n_c$ denotes the number of samples at client c.
>
> We will incorporate this clarification in the final paper.
>
> ---
>
> **Q4: Since only select prompts most similar to global prompts for aggregation, will this lead to a fixed prompts being aggregated while others always keep local? How is the frequency that each group prompt being aggregated globally?**
>
> **A4**: Thank you for this important question. Our method does not result in the same groups being consistently aggregated while others remain perpetually local. We provide aggregation frequency statistics for each group during training on OxfordPets in the table below. As mentioned in Q2, we introduce **a temperature parameter** to mitigate such extreme scenarios . Furthermore, our **dynamic aggregation mechanism** inherently requires different groups to maintain varying aggregation frequencies: prompt groups that are selected more frequently during training tend to aggregate **broader, global semantics**, while those with lower selection frequencies preserve more **localized, client-specific semantics** . These complementary behaviors balance generalization and personalization. Although we cannot include visualizations in this submission, we will add this discussion with qualitative examples and visualizations in the final version.
>
> | t_g1  | t_g2  | t_g3  | t_g4  | t_g5 | v_g1  | v_g2  | v_g3  | v_g4 | v_g5  |
> | ----- | ----- | ----- | ----- | ---- | ----- | ----- | ----- | ---- | ----- |
> | 28.0% | 17.0% | 23.5% | 12.5% | 19%  | 30.0% | 18.0% | 24.5% | 9.5% | 18.0% |

---

> > ### Author Response · Authors · 2025-08-04
> >
> > Dear Reviewer:
> >
> > We would like to ask if our response has fully addressed the concerns you raised. If there are any lingering questions or additional feedback you wish to share, please feel welcome to reach out to us directly. Thank you for your time.

---

> > > ### Comment · Reviewer_XV8h · 2025-08-05
> > >
> > > Thanks for the author's response, my concern has been resolved, I will maintain my positive score

---

> > > > ### Author Response · Authors · 2025-08-08
> > > >
> > > > Thank you for your reply. We sincerely appreciate your positive feedback on our paper and rebuttal, as well as your valuable suggestions for improving our paper.

---

### Official Review · Reviewer_t5MR · 2025-06-30

**Clarity:** 2
**Significance:** 2
**Originality:** 2
**Rating:** 4
**Confidence:** 3

**Summary:**

This paper proposes FedMGP, a personalized federated learning framework based on multi-group text-visual prompts, designed to address client overfitting and aggregation instability in vision-language models (VLMs) under heterogeneous data. The core contributions include:
1.Multi-group text-visual prompt pairs: each client maintains multiple paired text and visual prompts to enhance fine-grained semantic representation.
2.Diversity loss enforces complementary feature learning across different prompt groups.
3.Dynamic prompt aggregation utilizes similarity-based probabilistic sampling to balance global shared knowledge and client-specific information.

**Questions:**

Please refer to the weaknesses.

**Ethical Concerns:**

["NO or VERY MINOR ethics concerns only"]

**Final Justification:**

The response to my questions and discussion addressed my questions. I will raise my score.

**Limitations:**

Yes.

**Paper Formatting Concerns:**

No paper formatting concern.

**Quality:**

2

**Strengths And Weaknesses:**

Strengths:
1. The proposed method effectively addresses the expressiveness limitations of unimodal prompts. The dynamic aggregation strategy (similarity-weighted sampling) provides a compelling solution to the classic trade-off between personalization and generalization.
2. The proposed method is comprehensively evaluated across five datasets, Dirichlet-partitioned heterogeneous settings (CIFAR-10/100), and multi-domain scenarios (DomainNet), demonstrating strong generalizability.
3.FedMGP significantly reduces communication overhead compared to baselines, making it well-suited for resource-constrained federated environments.

Weaknesses:
1. The use of a fixed number of prompt groups (e.g., G=5) across all clients fails to consider data heterogeneity. Clients with simpler data (e.g., only 1–2 classes) may require fewer groups, while more complex clients (e.g., multi-domain or long-tail data) may benefit from more.
2.  The dynamic aggregation process relies on the cosine similarity between local prompts and the global prompts from the previous round. In the extreme case where a local prompt directly adopts the previous global prompt, the similarity reaches its maximum, resulting in the highest selection probability. However, this similarity-based sampling strategy may not be optimal.
3. At inference time, the model averages predictions across all prompt groups equally, ignoring confidence differences between groups. This uniform treatment could reduce prediction reliability, especially when some groups are more semantically aligned with the input than others.

---

> ### Author Rebuttal · Authors · 2025-07-30
>
> **Comment:**
>
> Thanks a lot for your time and feedback. We have to say that the reviewer asks valuable questions and provides thoughtful clues. We appreciate your inspiring reviews. And we are happy to address the concerns.
>
> ---
>
> **Q1: The use of a fixed number of prompt groups (e.g., G=5) across all clients fails to consider data heterogeneity. Clients with simpler data (e.g., only 1–2 classes) may require fewer groups, while more complex clients (e.g., multi-domain or long-tail data) may benefit from more.**
>
> **A1:** Thank you for this insightful question.
>
> - The referred setting corresponds to the Dirichlet distribution experiments in **Table 2 and Appendix D (α=0.5 / α=0.3)**, a widely adopted benchmark for measuring data heterogeneity[1][2]. Even under severe class imbalance, FedMGP delivers outstanding performance, demonstrating the robustness and generalization of our dynamic aggregation mechanism across diverse client distributions.
> - Moreover, according to your valuable suggestion. We have supplemented **an ablation study on the number of prompt groups (G) under this imbalanced setting**. The results are as follows. These findings show that G=5 strikes the optima**l balance between performance gains and parameter efficiency**, further enabled by our dynamic selection strategy that **adapts to each client’s data distribution**. Overall, FedMGP exhibits high efficiency, strong robustness, and superior generalization in handling data imbalance and heterogeneity.
>
> | **Group** | **cifar10** | **cifar100** |
> | --------- | ----------- | ------------ |
> | 2         | 93.36       | 72.27        |
> | 3         | 94.29       | 73.62        |
> | 4         | 94.35       | 74.57        |
> | 5         | **95.48**   | **75.39**    |
>
> [1] Li, Hongxia, et al. "Global and local prompts cooperation via optimal transport for federated learning." *Proceedings of the IEEE/CVF Conference on Computer Vision and Pattern Recognition*. 2024.
>
> [2] Cui, Tianyu, et al. "Harmonizing Generalization and Personalization in Federated Prompt Learning." International Conference on Machine Learning. PMLR, 2024.
>
> ---
>
> **Q2：The dynamic aggregation process relies on the cosine similarity between local prompts and the global prompts from the previous round. In the extreme case where a local prompt directly adopts the previous global prompt, the similarity reaches its maximum, resulting in the highest selection probability. However, this similarity-based sampling strategy may not be optimal.**
>
> **A2:** Thank you for you question. Our Method  have considered this scenario.
>
> - Our **dynamic aggregation** strategy incorporates a **temperature parameter (τ)** , which critically controls the selection probability distribution. As detailed in our ablation study in **Appendix E.1** , this parameter allows us to fine-tune the balance between **exploitation** (deterministically selecting high-similarity prompts with a low τ) and **exploration** (enabling the selection of diverse, lower-similarity prompts with a higher τ).
> - This **probabilistic sampling** , governed by the temperature, ensures a diverse set of prompts is aggregated in each round. **As demonstrated in the selection statistics table below** , even prompt groups with lower initial similarity are consistently given a chance to be selected across different communication rounds, effectively preventing premature convergence to a homogenized global prompt and fostering the continuous discovery of **diverse semantic features** .
> - Ultimately, this mechanism creates an intelligent trade-off inherent to personalized federated learning. Prompts frequently selected across clients tend to encapsulate **generalizable knowledge** , strengthening the core model, while less frequently selected prompts often hold unique, **client-specific features** crucial for **personalization** . Our method dynamically balances these two competing objectives, achieving a robust equilibrium between cross-client generalization and local adaptation.
> | round | t_g1 | t_g2 | t_g3 | t_g4 | t_g5 | v_g1 | v_g2 | v_g3 | v_g4 | v_g5 |
> |---|---|---|---|---|---|---|---|---|---|---|
> | 1 | 3(15%) | 2(10%) | 6(30%) | 5(25%) | 4(20%) | 3(15%) | 2(10%) | 6(30%) | 5(25%) | 4(20%) |
> | 2 | 6(30%) | 2(10%) | 6(30%) | 3(15%) | 3(15%) | 6(30%) | 2(10%) | 6(30%) | 3(15%) | 3(15%) |
> | 3 | 7(35%) | 2(10%) | 5(25%) | 3(15%) | 3(15%) | 7(35%) | 2(10%) | 5(25%) | 3(15%) | 3(15%) |
> | 4 | 6(30%) | 3(15%) | 5(25%) | 3(15%) | 3(15%) | 7(35%) | 3(15%) | 4(20%) | 2(10%) | 4(20%) |
> | 5 | 6(30%) | 3(15%) | 3(15%) | 3(15%) | 5(25%) | 7(35%) | 3(15%) | 3(15%) | 2(10%) | 5(25%) |
> | 6 | 6(30%) | 4(20%) | 3(15%) | 2(10%) | 5(25%) | 7(35%) | 4(20%) | 3(15%) | 1(5%) | 5(25%) |
> | 7 | 7(35%) | 4(20%) | 4(20%) | 1(5%) | 4(20%) | 7(35%) | 4(20%) | 4(20%) | 1(5%) | 4(20%) |
> | 8 | 6(30%) | 4(20%) | 4(20%) | 2(10%) | 4(20%) | 7(35%) | 4(20%) | 5(25%) | 1(5%) | 3(15%) |
> | 9 | 4(20%) | 5(25%) | 6(30%) | 1(5%) | 4(20%) | 5(25%) | 5(25%) | 7(35%) | 0(0%) | 3(15%) |
> | 10 | 5(25%) | 5(25%) | 5(25%) | 2(10%) | 3(15%) | 4(20%) | 7(35%) | 6(30%) | 1(5%) | 2(10%) |
>
> ---
>
> **Q3: At inference time, the model averages predictions across all prompt groups equally, ignoring confidence differences between groups. This uniform treatment could reduce prediction reliability, especially when some groups are more semantically aligned with the input than others.**
>
> **A3:** We appreciate your valuable comments. We wish to clarify that our inference strategy directly incorporates the confidence of each group, as you suggested. For each prompt group, the **final logits** , computed from the similarity between its guided text and image features, directly **represent that group's confidence distribution over the classes for a given input**. Therefore, averaging these logits is a principled way to ensemble the predictions, where each group's contribution is naturally weighted by its confidence. As rigorously demonstrated in our **ablation study in Table 6** , this method significantly outperforms alternatives like selecting the single most confident group (Max logits) or averaging the prompt-guided features. This result powerfully validates our integrated design: the multi-group architecture enables diverse specializations—some capturing broad, generalizable knowledge while others master **client-specific, personalized nuances** . Our inference strategy then optimally synthesizes these varied perspectives, allowing the model to draw upon shared knowledge for general tasks while accurately reflecting personal characteristics when needed, thus achieving a superior balance between **generalization and personalization** .

---

> ### Author Response · Authors · 2025-08-04
>
> Dear Reviewer:
>
> We would like to ask if our response has sufficiently addressed your concerns. If you have any remaining questions or concerns, please feel free to share with us directly. Thank you for your time.

---

> ### Comment · Reviewer_t5MR · 2025-08-05
>
> Thank you for your response. For Q2, although more justification is provided for dynamic aggregation and probabilistic sampling, they cannot address the question about the design of the similarity evaluation. “In the extreme case where a local prompt directly adopts the previous global prompt, the similarity reaches its maximum, resulting in the highest selection probability.” A more reasonable probability design is needed.

---

> ### Author Response · Authors · 2025-08-05
>
> Thank you for your feedback. We understand your concerns and would like to further clarify the rationality of the similarity design. The detailed explanation is as follows:
>
> - We would like to clarify that in the federated learning process(Appendix A), **there is no scenario where a local prompt directly adopts the previous global prompt when aggregatting .** After aggregation, **only the top-s groups use the global prompt for initialization, but after local training in each client, they incorporate a substantial amount of local knowledge, resulting in significant differences from the previous round's global prompt when aggregation in current round.**
> - We understand your concern that the initialized groups with previous round's global prompt might always maintain the highest similarity, leading to the exclusion of other groups. However, in practice:
>   - Due to the diversity loss, **the initialized prompts will pull away from each other during local training**, learning diverse features and ensuring that each group has the opportunity to **capture general knowledge** (as confirmed in **Appendix D**) .
>   - Due to local data heterogeneity, in federated learning, **client data is highly heterogeneous (such as Dirichlet distribution)**, causing the prompts after local training to differ greatly from their initial values, **preventing situations where similarity to the previous round's global prompt is extremely high.**
>   - The temperature parameter (τ) introduces randomness, allowing **low-similarity prompts (containing more local features) to participate in aggregation at a low frequency, balancing exploration and exploitation**, preventing overfitting, and minimizing interference (as shown in the ablation study in **Appendix E.1**) .
>   - The following table shows the similarity between each group's local prompt and the previous round's global prompt before and after each round of training for the OxfordPets dataset, cilent1. It can be seen that **after local training, each group will have significant differences from the initial group, especially the group which parameters were updated by the previous round's global prompt.** The bolded entries in the table represent the group with the highest similarity in each round, and * represents the group selected in the current round. **Due to the presence of the temperature parameter, the group with the highest similarity is not always selected (e.g., round4, round6). The group with the highest similarity each time is not necessarily the group which parameters were initialized with the previous round's global prompt (e.g., round2, round4).**
>
> | **Round**        | **group1** | **group2** | **group3** | **group4** | **group5** |
> | ---------------- | ---------- | ---------- | ---------- | ---------- | ---------- |
> | **round2_pre**   | 0.67       | **1.00\*** | **1.00\*** | 0.56       | 0.49       |
> | **round2_after** | **0.63\*** | 0.41       | **0.60\*** | 0.57       | 0.43       |
> | **round3_pre**   | **1.00\*** | 0.39       | **1.00\*** | 0.35       | 0.49       |
> | **round3_after** | **0.72\*** | 0.41       | **0.51\*** | 0.34       | 0.42       |
> | **round4_pre**   | **1.00\*** | 0.58       | **1.00\*** | 0.31       | 0.67       |
> | **round4_after** | **0.69**   | 0.53       | 0.51*      | 0.33*      | **0.61**   |
> | **round5_pre**   | 0.69       | 0.64       | **1.00\*** | **1.00\*** | 0.66       |
> | **round5_after** | **0.64\*** | 0.61       | 0.33       | 0.38       | **0.62\*** |
> | **round6_pre**   | **1.00\*** | 0.63       | 0.42       | 0.20       | **1.00\*** |
> | **round6_after** | **0.67\*** | 0.62*      | 0.40       | 0.21       | **0.68**   |
> | **round7_pre**   | **1.00\*** | **1.00\*** | 0.60       | 0.23       | 0.51       |
> | **round7_after** | **0.80\*** | **0.62**   | 0.62       | 0.22*      | 0.50       |
> | **round8_pre**   | **1.00\*** | 0.56       | 0.74       | **1.00\*** | 0.37       |
> | **round8_after** | **0.70\*** | 0.57       | **0.76**   | 0.49       | 0.36*      |
> | **round9_pre**   | **1.00\*** | 0.53       | 0.81       | 0.44       | **1.00\*** |
> | **round9_after** | 0.56       | **0.61**   | **0.84\*** | 0.41       | 0.39*      |

---

> > ### Author Response · Authors · 2025-08-05
> >
> > - Moreover, our similarity design also has been validated through theoretical proof and experimental verification. The similarity is not designed "for similarity's sake," but **serves as an effective proxy for high SNR (Signal-to-Noise Ratio) (as proven in Theorem F.1, Appendix F)** . High-similarity prompts represent the learning of more **common features rather than local bias features**, promoting the aggregation of universal features, reducing heterogeneous interference, and achieving an optimal **balance between personalization and generalization.**
> > - This design has been verified through many experiments, outperforming fixed aggregation under heterogeneous data **(Table 1-2, Appendix D.1, Appendix D.2)** . It compares the **similarity-based dynamic aggregation strategy** with other fixed aggregation strategies. The ablation experiments for the dynamic aggregation mechanism (DAM) are shown in the following table (averaged over 5 datasets), which further demonstrates that our DAM, compared to the complete aggregation strategy (CAM) or the fixed aggregation mechanism (FAM), **can better balance generalization and personalization.**
> >
> > | Method    | Local | Base  | Novel | CM    |
> > | --------- | ----- | ----- | ----- | ----- |
> > | CAM       | 86.13 | 85.94 | 83.57 | 85.43 |
> > | FAM       | 97.37 | 77.47 | 78.71 | 87.73 |
> > | DAM(ours) | 96.65 | 79.2  | 80.86 | 88.34 |
> >
> > **If you have any remaining questions or concerns, please feel free to share with us directly what specific points are on your mind.**

---

> > > ### Author Response · Authors · 2025-08-08
> > >
> > > Dear Reviewer,
> > >
> > > We would like to kindly inquire if our response has addressed your concerns regarding Q2. If you have any further questions or points of uncertainty, we would appreciate it if you could let us know. We eagerly await your feedback.

---

### Official Review · Reviewer_8EoJ · 2025-07-01

**Clarity:** 3
**Significance:** 2
**Originality:** 3
**Rating:** 4
**Confidence:** 3

**Summary:**

This paper presents FedMGP, a personalized federated prompt learning framework for VLMs. Their proposed approach is designed to involve both viusal and text prompt, which is different from single textual prompts and static local-global aggregation. In the proposed system, multi-group text-visual prompt co-learning at each client to capture diverse semantic and instance-specific features, where a diversity loss to encourage representation separation across prompt groups. A similarity based dynamic prompt aggregation strategy is proposed to balance personalization and generalization. The authors compared to PromptFL, FedOPT, FedTPG, FedPGP and PromptFolio across datasets including CIFAR, Flowers, DTD, Caltech101, etc under heterogeneity settings.

**Questions:**

Questions:
1. In many group-based FL algorithms, group membership tends to drift across rounds. Could the authors clarify whether prompt group assignments remain stable across training rounds? Is there any tracking or visualization of how group semantics evolve over time?
2. Fairness and bias: Do the multi-group prompts risk encoding biased semantic clusters, especially if clients have class-imbalanced datasets?
3. The diversity loss encourages group-wise differentiation, but it remains unclear what kinds of semantics each group learns. Could the authors provide qualitative examples or visualizations?

**Ethical Concerns:**

["NO or VERY MINOR ethics concerns only"]

**Final Justification:**

Final justification

**Limitations:**

yes

**Paper Formatting Concerns:**

- Table 4 should leangth -> length
- Line 96, "we introduces" -> "we introduce"
- Line 82, multiple of [38]

**Quality:**

3

**Strengths And Weaknesses:**

Strength:
- The paper proposed multi-group text-visual prompt design, enabling richer client-specific representation compared to single-text prompt baselines. The softmax-based prompt group selection balances global knowledge sharing and local personalization under data heterogeneity
- The authors evaluated across various datasets under non-IID settings; consistently achieves state-of-the-art results in both personalization and generalization.  Validates the impact of prompt length, group count, aggregation strategy, modality contributions, and diversity loss.
Weakness:
- The paper does not report error bars or statistical significance tests, which limits the strength of its claims regarding performance robustness under data variability.
- The paper claims that different prompt groups specialize in distinct semantics via diversity loss, but provides less interpretability and explainability.
- Although the method is motivated by privacy-sensitive federated settings, the paper does not assess whether the shared prompt vectors, especially visual prompts could leak private information or be vulnerable to reconstruction attacks.

---

> ### Author Rebuttal · Authors · 2025-07-30
>
> **Comment:**
>
> We appreciate your valuable comments. We were wondering if our responses have addressed your concerns. Please let us know if you have additional questions. Thank you!
>
> ---
>
> **Q1:The paper does not report error bars or statistical significance tests, which limits the strength of its claims regarding performance robustness under data variability.The paper does not report error bars or statistical significance tests, which limits the strength of its claims regarding performance robustness under data variability.**
>
> **A1**: To rigorously evaluate performance robustness, we have conducted main experiments across the five datasets using **three distinct random seeds (0, 1, 2)** . The updated table presents the results, confirming that FedMGP not only achieves the **highest performance** but also demonstrates **superior stability** with the lowest standard deviation. Crucially, this leading performance and robustness are achieved with the **lowest communication overhead (2.6k parameters)** among all methods, highlighting the exceptional efficiency and reliability of our approach.
>
> | **Method** | **OxfordPets** | **Flowers102** | **DTD** | **Caltech101** | **Food101** |
> |-------------|----------------|----------------|--------------|----------------|--------------|
> | PromptFL    | 94.39±1.45     | 80.74±0.09     | 64.53±1.98   | 95.25±1.13     | 88.91±0.47   |
> | FedOTP      | 63.83±2.38     | 60.03±0.66     | 62.11±2.50   | 73.59±4.31     | 59.68±1.69   |
> | FedTPG      | 95.22±1.03     | 79.12±1.74     | 61.21±1.68   | 96.11±0.41     | 90.61±0.22   |
> | FedPGP      | 95.77±0.45     | 82.63±0.76     | 67.42±2.47   | 96.18±0.29     | 90.45±0.25   |
> | PromptFolio | 77.58±8.54     | 70.11±1.71     | 62.17±1.60   | 89.19±0.45     | 79.82±1.32   |
> | **FedMGP** | **95.99±0.11** | **84.87±0.96** | **72.42±0.50** | **96.98±0.21** | **91.03±0.74** |
>
> ---
>
> **Q2: The paper claims that different prompt groups specialize in distinct semantics via diversity loss, but provides less interpretability and explainability.**
>
> **A2:**  Thank you for this suggestion. We fully recognize the importance of interpretability and have addressed it in our paper as follows:
>
> - As shown in Figure 1 of the appendix, our diversity loss effectively enforces distinct representations. The **remarkably low inter-group correlation** among visual prompts provides direct evidence that each group learns to capture unique and complementary visual features.
> - In appendix F, **signal-noise decomposition theory** mathematically proves that the multi-group mechanism enhances the model's **signal-to-noise ratio** by promoting representational diversity . This framework provides a rigorous foundation for why our design choice is critical for balancing personalization and generalization.
>
> ---
>
> **Q3: Although the method is motivated by privacy-sensitive federated settings, the paper does not assess whether the shared prompt vectors, especially visual prompts could leak private information or be vulnerable to reconstruction attacks.**
>
> **A3**: We appreciate the reviewer’s concern. Our framework adheres to the **FedAvg paradigm[1]**, which has been extensively proven to offer strong privacy protection because:
>
> - Our method exchanges only updated text and visual prompt parameters, **without sharing gradients, prediction logits, or data embeddings**, eliminating the key information required for model inversion, and preventing adversaries from obtaining labels or embeddings to build shadow models.
> - With only approximately **2.6K parameters** per round, PromptFL[2] has shown that low-parameter communication is more privacy-preserving than full model uploads; this also drastically reduces the attack surface and minimizes the risk of membership inference and attribute inference.
> - Furthermore, our method leverages diversity loss and  dynamic selection to prevent overfitting of single prompts and enhance protection of privacy-sensitive features.
>
> [1] McMahan, Brendan, et al. "Communication-efficient learning of deep networks from decentralized data." *Artificial intelligence and statistics*. PMLR, 2017.
>
> [2]Guo, Tao, et al. "Promptfl: Let federated participants cooperatively learn prompts instead of models–federated learning in age of foundation model." *IEEE Transactions on Mobile Computing* 23.5 (2023): 5179-5194.
>
> ---
>
> **Q4:In many group-based FL algorithms, group membership tends to drift across rounds. Could the authors clarify whether prompt group assignments remain stable across training rounds? Is there any tracking or visualization of how group semantics evolve over time?**
>
> **A4**: Thank you for your suggestion! Our prompt group assignments remain highly stable throughout training.  We quantitatively demonstrate this by **tracking each group’s selection frequency per round.**
>
> - As shown in the table below, which demonstrates the selection frequency of each group during training on the OxfordPets dataset. Our **dynamic aggregation mechanism** is designed to guide the specialization of prompts; some evolve to capture **shared global knowledge** by being selected frequently, while others focus on **client-specific local features** through less frequent selection.
> - The **temperature parameter (τ)** in our selection process is crucial for maintaining this dynamic balance, preventing any prompt group from being permanently excluded and thereby ensuring the model retains both **generalization and personalization** capabilities.
> | round | t_g1 | t_g2 | t_g3 | t_g4 | t_g5 | v_g1 | v_g2 | v_g3 | v_g4 | v_g5 |
> |---|---|---|---|---|---|---|---|---|---|---|
> | 1 | 3(15%) | 2(10%) | 6(30%) | 5(25%) | 4(20%) | 3(15%) | 2(10%) | 6(30%) | 5(25%) | 4(20%) |
> | 2 | 6(30%) | 2(10%) | 6(30%) | 3(15%) | 3(15%) | 6(30%) | 2(10%) | 6(30%) | 3(15%) | 3(15%) |
> | 3 | 7(35%) | 2(10%) | 5(25%) | 3(15%) | 3(15%) | 7(35%) | 2(10%) | 5(25%) | 3(15%) | 3(15%) |
> | 4 | 6(30%) | 3(15%) | 5(25%) | 3(15%) | 3(15%) | 7(35%) | 3(15%) | 4(20%) | 2(10%) | 4(20%) |
> | 5 | 6(30%) | 3(15%) | 3(15%) | 3(15%) | 5(25%) | 7(35%) | 3(15%) | 3(15%) | 2(10%) | 5(25%) |
> | 6 | 6(30%) | 4(20%) | 3(15%) | 2(10%) | 5(25%) | 7(35%) | 4(20%) | 3(15%) | 1(5%) | 5(25%) |
> | 7 | 7(35%) | 4(20%) | 4(20%) | 1(5%) | 4(20%) | 7(35%) | 4(20%) | 4(20%) | 1(5%) | 4(20%) |
> | 8 | 6(30%) | 4(20%) | 4(20%) | 2(10%) | 4(20%) | 7(35%) | 4(20%) | 5(25%) | 1(5%) | 3(15%) |
> | 9 | 4(20%) | 5(25%) | 6(30%) | 1(5%) | 4(20%) | 5(25%) | 5(25%) | 7(35%) | 0(0%) | 3(15%) |
> | 10 | 5(25%) | 5(25%) | 5(25%) | 2(10%) | 3(15%) | 4(20%) | 7(35%) | 6(30%) | 1(5%) | 2(10%) |
>
> ---
>
> **Q5:Fairness and bias: Do the multi-group prompts risk encoding biased semantic clusters, especially if clients have class-imbalanced datasets?**
>
> **A5**: Thank you for your valuable Question. **Our FedMGP method demonstrates superior generalization and carries no risk of encoding biased semantic clusters.** In the main paper (Table 2, Dirichlet α=0.5), we show exceptional performance under severe class imbalance . In the cross-domain experiments in Appendix D, we apply an even stricter Dirichlet distribution (α=0.3) on each domain; **even under simultaneous feature and label shifts, FedMGP maintains robust performance**, directly validating the effectiveness of our dynamic aggregation mechanism.
>
> ---
>
> **Q6：The diversity loss encourages group-wise differentiation, but it remains unclear what kinds of semantics each group learns. Could the authors provide qualitative examples or visualizations?**
>
> **A6**: Thank you for the suggestion. As demonstrated in the Q2 experiments, **prompt groups that are selected more frequently during training tend to aggregate broader, global semantics, while those with lower selection frequencies preserve more localized, client-specific semantics.** These complementary behaviors balance generalization and personalization. Although we cannot include visualizations in this submission, we will add this discussion with qualitative examples and visualizations in the final version.

---

> > ### Author Response · Authors · 2025-08-04
> >
> > Dear Reviewer:
> >
> > We would like to inquire whether our response has adequately addressed your concerns. If you have any further questions or remaining concerns, please do not hesitate to let us know directly. Thank you for your time.

---

### Note · Authors · 2025-08-13

We sincerely appreciate the valuable feedback from all reviewers, which has helped us refine our work. **For convenience, we summarize the key review questions and our response.**

| **Key Comment/Question**                                     | **Response**                                                 |
| ------------------------------------------------------------ | ------------------------------------------------------------ |
| **Reviewer 8EoJ**                                            |                                                              |
| good design; excellent effect; thorough experiments          | We thank Rev. 8EoJ for the positive feedback.                |
| no statistical significance tests; limited explainability of group semantics; | We added statistical significance tests in Q1; provided detailed explanations of the prompt group in **Q2, Q5, and Q6. The reviewer did not raise further concerns.** |
| **Reviewer t5MR**                                            |                                                              |
| addresses the expressiveness limitations; dynamic aggregation strategy balances the classic trade‑off | We are encouraged by Rev. t5MR's recognition of our key contributions. |
| extreme situations in federated learning with dynamic aggregation | We provided detailed experimental data and theoretical explanations showing that FedMGP can effectively handle these situations **(Q2 & Official Comment on 05 Aug). The reviewer did not raise further concerns.** |
| **Reviewer XV8h**                                            |                                                              |
| interesting design; competitive performance                  | We’re grateful to Rev.XV8h for the positive remarks.         |
| lack of discussion on related group prompt; dynamic prompt aggregation in special situations | We addressed these issues in **Q1, Q2, and Q4, and the reviewer considered all concerns resolved.** |
| **Reviewer KEQK**                                            |                                                              |
| method is novel and well‑motivated; comprehensive experiments and theoretical analysis | We appreciate Rev. KEQK’s acknowledgment of our contributions. |
| more methods and datasets for comparison; reason for generalization  with prompt diversity | We added relevant experiments in Q1 and Q4; after multiple discussions, t**he reviewer agreed that all concerns were addressed.** |

---

### Decision · Program_Chairs · 2025-09-17

**Decision:**

Accept (poster)

**Comment:**

This study presents a personalized Federated Learning framework developed to alleviate client overfitting and aggregation instability in VLMs by utilizing a multi-group text-visual prompt design, a diversity loss and dynamic prompt aggregation based on similarity probabilistic sampling to balance global and client-specific knowledge. All reviewers recognised that the proposed approach is meaningful and novel, and it is sufficiently evaluated over a decent number of benchmarks (under non-IID settings). At the same time, they identified a number of technical issues which I believe that the authors have sufficiently addressed during the rebuttal (as also recognized by the majority of the reviewers). Since (almost all of) the concerns are resolved, I'm happy to recommend accept for this paper.